

# Evaluation of equivalent black carbon (EBC) source appointment using observations from Switzerland between 2008 and 2018

Stuart K. Grange[1,2], Hanspeter Lötscher[3], Andrea Fischer[1], Lukas Emmenegger[1], and
Christoph Hueglin[1]

[1]Empa, Swiss Federal Laboratories for Materials Science and Technology, Überlandstrasse 129, 8600 Dübendorf, Switzerland
[2]Wolfson Atmospheric Chemistry Laboratories, University of York, York, YO10 5DD, United Kingdom
[3]Amt für Natur und Umwelt Graubünden, Gürtelstrasse 89, 7001 Chur, Switzerland

*Correspondence to:* Stuart K. Grange (stuart.grange@empa.ch)

**Abstract.**

Black carbon (BC) or soot is a constituent of particulate matter (PM) which is relevant for negative human health and climate effects, and despite the lack of legal limits, it is recognised as an important atmospheric pollutant to monitor, understand, and control. Aethalometers are instruments which continuously monitor BC by measuring absorption at a number of distinct
wavelengths. If collocated elemental carbon (EC) observations are used to transform these values into BC mass, by convention, the result is named equivalent black carbon (EBC). BC emitted by different combustion processes have different optical absorption characteristics and this can be used to apportion EBC mass into traffic ($EBC_{TR}$) and woodburning ($EBC_{WB}$) components with a data processing technique known as the aethalometer model. The aethalometer model was applied to six EBC monitoring sites across Switzerland (using data between 2008 and 2018) and was evaluated by investigating diurnal cycles,
model coefficients, and ambient temperature dependence of the two EBC components. For one monitoring site, San Vittore, the aethalometer model failed to produce plausible outputs. The reason for this failure was likely due to a high load of freshly emitted wood smoke during the winter which should be thought of as a third distinct emission source. After model evaluation, the trend analysis indicated that $EBC_{TR}$ concentrations at the remaining five locations significantly decreased between 2008 and 2018 illustrating the successful widespread installation of diesel particulate filters (DPF) within the vehicle fleet. $EBC_{WB}$
also demonstrated significant deceases in most monitoring locations, but not at a monitoring site south of the Alps with a high PM load sourced from biomass burning. This indicated that the management of woodburning has be ineffective at reducing BC emissions and concentrations for this, likely representative location. The $EBC/PM_{2.5}$ ratios suggested that EBC contributes 6–13 % of the $PM_{2.5}$ mass in Switzerland which is important for soot and PM source management. The aethalometer model is a useful data analysis procedure, but can fail under certain conditions, thus, careful evaluation is required to ensure the method
is robust and suitable in other locations.





# 1  Introduction

## 1.1  Black carbon

Atmospheric particulate matter (PM) has a variety of components, one of which is black carbon (BC) which is commonly referred to as soot. BC is strongly light absorbing (hence the name), is generally found in the fine PM fraction ($\leq PM_{2.5}$),

and is generated by the incomplete combustion of fuels (Hansen et al., 1984; Vignati et al., 2010). Biomass burning, internal combustion engines (especially those which are fuelled by diesel), and industrial processes can all be BC emission sources (Bond et al., 2004; Jacobson, 2001). There are no known BC generation or degradation processes in the atmosphere and because the only removal mechanisms for BC are wet and dry deposition, BC has a lifetime between several days to weeks in the atmosphere. These attributes make BC a reliable tracer for combustion processes.

BC is an important atmospheric pollutant to consider in its own sense however (Anenberg et al., 2012). BC is believed to have a positive (warming) climate forcing effect due to the absorption of radiation and the reduction of albedo, especially when deposited on snow and ice (Ramanathan and Carmichael, 2008; Weinhold, 2012; Bond et al., 2013). BC is harmful to human health and there is evidence that BC is a particularly potent PM component when considering deleterious health effects due to the aerosol's large surface area to mass ratio, tendency to be enriched with harmful organic compounds, and ability to

penetrate deep into the lungs (Janssen et al., 2011; De Prins et al., 2014; Laeremans et al., 2018). The International Agency for Research on Cancer (IARC) has also classed soot as a Group 1 carcinogen indicating there is clear evidence that soot causes cancer in humans (International Agency for Research on Cancer, 2019). It is however, difficult to tease apart the health effects of BC alone from PM as a whole (Jacobson, 2001; European Environment Agency, 2013). These features have made BC an important pollutant to monitor, understand, manage, and control despite not currently having legal limits imposed in Europe

and elsewhere (Reche et al., 2011; European Environment Agency, 2016).

## 1.2  Aethalometers and the aethalometer model

BC is measured, sampled, or monitored in a variety of ways but the most widespread is done by aethalometers (Hansen et al., 1984). Aethalometers are described in depth elsewhere, however, briefly they are optical instruments which sample air continuously and deposit the PM onto filter material. The loaded filter is illuminated, the light attenuation is measured as an

optical absorption, and the increase in attenuation over time is logged and used to calculate BC concentrations with empirical coefficients. The quartz filter material used by aethalometers can reach saturation, so the instruments employ a filter-tape system where the filter material is advanced if saturation occurs, or when a threshold of elapsed time has passed. Continuously depositing aerosol onto a filter causes artefacts known as filter-scattering and filter-shadowing effects, but there are standard algorithms to compensate for these effects, some of which are applied on-board by the instrument as part of the measurement

cycle (Magee Scientific, 2016).

Modern aethalometers used in ambient air quality monitoring networks are multi-wavelength models measuring absorption between the near-ultraviolet (UV) to the near-infrared (IR) range, but at distinct wavelengths within this range. If the wavelength range is wide enough, analysis of the spectral dependence can be explored. BC sourced from different families of combustion





processes demonstrate different absorption dependence at different wavelengths. Most notably, woodburning particles, *i.e.* wood smoke, generally contains a rich organic component which is very effective at absorbing light in the UV range while diesel soot shows a weaker spectral dependence (Kirchstetter et al., 2004) (Fig. 1). The different spectral dependences between these sources can be leveraged with a data processing technique called the aethalometer model, first reported by Sandradewi et al. (2008) but used many times since in many locations; for examples, see Sandradewi et al. (2008); Favez et al. (2009); Herich et al. (2011); Fuller et al. (2014). In routine aethalometer use, only the 880 nm absorption wavelength is used with a fixed coefficient (called a mass absorption coefficient) which is the strict definition of BC. Therefore, the aethalometer model simply uses information which is generally discarded in most monitoring applications and the barrier for entry to the method is low.

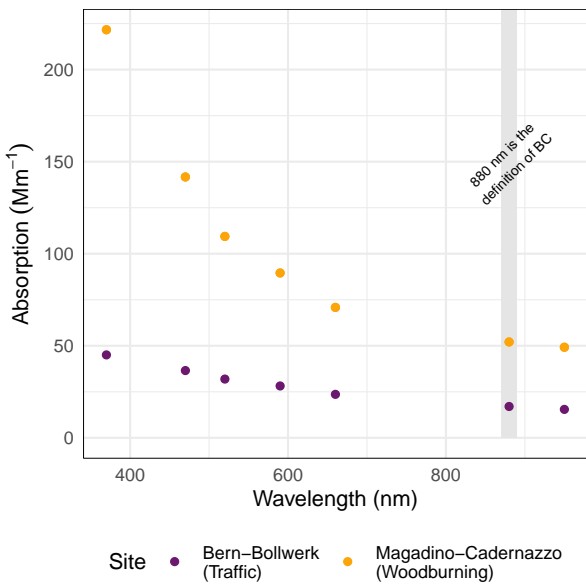

**Figure 1.** Demonstration of different aethalometer absorption dependence on wavelengths for two monitoring sites in Switzerland with distinct aerosol characteristics. Data have been filtered to a single observation (hourly) to show the dependence on the dominating sources.

## 1.3 Black carbon nomenclature note

The nomenclature and vocabulary used for BC is diverse and can lead to confusion (Andreae and Gelencsér, 2006; Petzold et al., 2013). Here, we refer to BC determined by optical measurement techniques, *i.e.* by aethalometers and transformed with the use of collocated elemental carbon (EC) observations, as equivalent black carbon (EBC). This definition has been recommended by the Global Atmospheric Watch (Tarasova, 2012). When EBC is apportioned into its traffic and woodburning components, the TR and WB subscript notation is used respectively, *i.e.*, $EBC_{TR}$ and $EBC_{WB}$. If total EBC needs to be used for clarity against the EBC components, $EBC_{TOT}$ is used. This notation is consistent with Zotter et al. (2017) which this





work somewhat extends. When discussing emissions of soot, BC is still used however, because EBC is only a measurement definition and does not refer to the pollutant.

## 1.4 Objectives

The primary objective of this work is to apply the aethalometer model data transformation technique to apportion long-term
(up to ten years) EBC time series from six Swiss monitoring sites into two components: a traffic and a woodburning component ($EBC_{TR}$ and $EBC_{WB}$) and evaluate the method's performance. The suitability and limitations of the aethalometer model as a technique to employ before trend analysis will be discussed. A data set is presented where the aethalometer model fails to apportion the two EBC components correctly. This is done to demonstrate the features which will be present if the aethalometer model is inadequate for a particular application, and it outlines that although the aethalometer model is a useful technique, but is
not a panacea. The second objective is to expose the $EBC_{TR}$ and $EBC_{WB}$ components to a trend analysis and give explanations for the features observed in the Swiss time series.

## 2 Methods

### 2.1 Data

Absorption observations measured by aethalometers for six monitoring sites in Switzerland were analysed (Table 1; Fig. 2).
The monitoring sites were either part of the federal monitoring network (National Air Pollution Monitoring Network; NABEL) or networks run by the Swiss Cantonal authorities (states) (Federal Office for the Environment, 2014). Sites were classified as either urban traffic, urban background, rural, or rural mountain according to their surrounds. Four of the sites were located on the Swiss plateau (where the majority of the human population is located) but Rigi-Seebodenalp, despite being located on the plateau is at altitude (1031 metres; Figure 2). Two monitoring sites, Magadino-Cadenazzo and San Vittore are located south
of the Alps in valleys where residential woodburning is much more common than on the Swiss plateau due to its rural nature (Alfarra et al., 2007; Szidat et al., 2007).

**Table 1.** Information about the six equivalent black carbon (EBC) monitoring sites in Switzerland which were used in this study.

| Site | Site name | Local ID | Site type | Start date | End date | Days online | Latitude | Longitude | Elevation (m) |
|---|---|---|---|---|---|---|---|---|---|
| ch0031a | Bern-Bollwerk | BER | Urban traffic | 2013-12-11 | 2018-12-31 | 1846 | 46.951 | 7.441 | 536 |
| ch0010a | Zürich-Kaserne | ZUE | Urban background | 2009-04-02 | 2018-12-31 | 3560 | 47.378 | 8.530 | 409 |
| ch0002r | Payerne | PAY | Rural | 2008-03-04 | 2018-12-31 | 3954 | 46.813 | 6.944 | 489 |
| ch0033a | Magadino-Cadenazzo | MAG | Rural | 2008-03-13 | 2018-12-31 | 3945 | 46.160 | 8.934 | 203 |
| ch2001e | San Vittore | SVI | Rural | 2013-10-04 | 2018-12-31 | 1914 | 46.239 | 9.105 | 298 |
| ch0005r | Rigi-Seebodenalp | RIG | Rural mountain | 2013-01-25 | 2018-12-31 | 2166 | 47.067 | 8.463 | 1031 |

The absorption observations were measured by two models of multi-wavelength aethalometers manufactured by Magee Scientific. The two models were the AE31 and AE33 which measure adsorption at seven wavelengths: 370, 470, 520, 590,





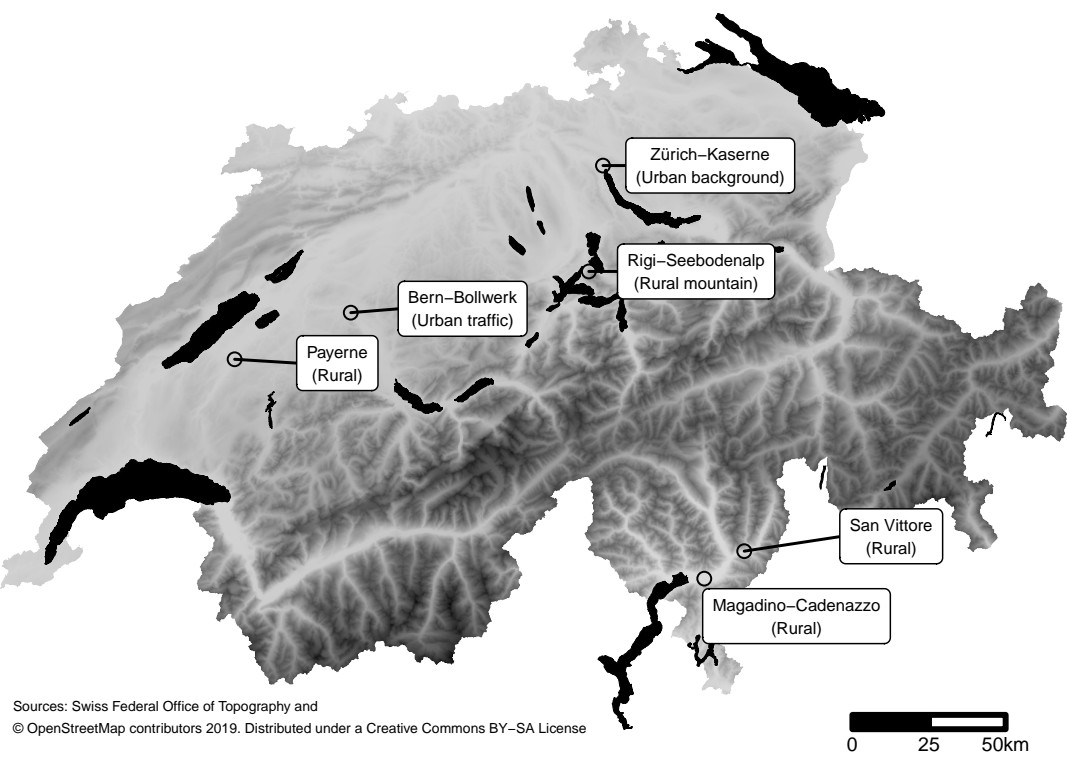

**Figure 2.** Switzerland's six equivalent black carbon (EBC) monitoring sites and their site classifications. Solid black areas are significant water bodies and shading represents the terrain and elevation. The boundaries of Switzerland and the lakes were extracted from OpenStreetMap (OpenStreetMap contributors, 2019) and the heights were derived from Switzerland's digital height model (Swiss Federal Office of Topography (swisstopo), 2010).

660, 880, and 950 nm. The AE31 is an earlier generation instrument discontinued in 2016 (Magee Scientific, 2017) while the AE33 aethalometer replaced the AE31 and is currently manufactured (Magee Scientific, 2019). These two aethalometer models share the same measurement principle, however the newer AE33 uses a "dual-spot" approach to allow for a superior method to compensate for filter loading effects (Drinovec et al., 2015). The differences between the AE31 and AE33 technologies mean

5    that the algorithms which compensate or correct for filter shadowing effects and filter loading effects are different. For the full description of the compensation procedures, see Weingartner et al. (2003); Drinovec et al. (2015). Quartz filters were used by the aethalometers and therefore, a multiple light scattering factor (denoted as $C$) of 2.14 was used (Weingartner et al., 2003). There were thirteen distinct aethalometers used in the monitoring network which were irregularly cycled among the monitoring sites as part of the networks' maintenance procedures (Table 2). All aethalometers were operated with $PM_{2.5}$ sample inlets.

10    Data from the aethalometers for five of the six Swiss EBC monitoring sites were queried from the NABEL monitoring network's database while the remaining site's observations, San Vittore, were provided directly by the Graubünden canton's environmental department (Amt für Natur und Umwelt). All absorption observations had been compensated for the filter loading and shadowing effects with the instrument model's respective algorithms before this analysis was undertaken (Weingartner





**Table 2.** Aethalometer instruments' locations between 2007 and 2018. Dates have been rounded to the nearest hour and those end dates which are missing shows the instrument was on site to the end of the analysis period (the end of 2018).

| Site | Site name | Local ID | Start date | End date | Instrument ID |
|------|-----------|----------|------------|----------|---------------|
| ch0031a | Bern-Bollwerk | BER | 2013-12-10 16:00:00 | 2014-03-13 10:00:00 | AE33-S02-00136 |
| ch0031a | Bern-Bollwerk | BER | 2014-03-13 09:00:00 | 2014-03-14 16:00:00 | AE33-S01-00092 |
| ch0031a | Bern-Bollwerk | BER | 2014-03-14 15:00:00 | 2016-07-27 11:00:00 | AE31-768:0701 |
| ch0031a | Bern-Bollwerk | BER | 2016-07-27 15:00:00 | 2016-10-26 10:00:00 | AE31-718:0605 |
| ch0031a | Bern-Bollwerk | BER | 2016-10-26 10:00:00 | 2017-07-06 11:00:00 | AE31-768:0701 |
| ch0031a | Bern-Bollwerk | BER | 2017-07-06 10:00:00 | 2018-03-22 12:00:00 | AE31-719:0605 |
| ch0031a | Bern-Bollwerk | BER | 2018-03-22 11:00:00 | 2018-11-29 12:00:00 | AE31-768:0701 |
| ch0031a | Bern-Bollwerk | BER | 2018-11-29 10:00:00 | | AE33-S07-00736 |
| ch0010a | Zürich-Kaserne | ZUE | 2009-04-02 10:00:00 | 2009-10-08 11:00:00 | AE31-769:0701 |
| ch0010a | Zürich-Kaserne | ZUE | 2009-10-08 10:00:00 | 2011-07-25 14:00:00 | AE31-768:0701 |
| ch0010a | Zürich-Kaserne | ZUE | 2011-07-25 13:00:00 | 2012-08-31 14:00:00 | AE31-719:0605 |
| ch0010a | Zürich-Kaserne | ZUE | 2012-08-31 10:00:00 | 2013-06-05 13:00:00 | AE31-768:0701 |
| ch0010a | Zürich-Kaserne | ZUE | 2013-06-05 12:00:00 | 2014-04-10 11:00:00 | AE31-718:0605 |
| ch0010a | Zürich-Kaserne | ZUE | 2014-04-10 10:00:00 | | AE33-S01-00092 |
| ch0002r | Payerne | PAY | 2008-03-04 15:00:00 | 2008-05-14 09:00:00 | AE31-768:0701 |
| ch0002r | Payerne | PAY | 2008-05-21 09:00:00 | 2008-10-02 16:00:00 | AE31-768:0701 |
| ch0002r | Payerne | PAY | 2008-10-02 15:00:00 | 2013-04-04 12:00:00 | AE31-718:0605 |
| ch0002r | Payerne | PAY | 2013-04-04 11:00:00 | 2013-06-27 13:00:00 | AE31-719:0605 |
| ch0002r | Payerne | PAY | 2013-06-27 12:00:00 | 2015-09-16 12:00:00 | AE31-769:0701 |
| ch0002r | Payerne | PAY | 2015-09-16 12:00:00 | 2016-06-08 16:00:00 | AE31-719:0605 |
| ch0002r | Payerne | PAY | 2016-06-08 10:00:00 | | AE33-S04-00430 |
| ch0033a | Magadino-Cadenazzo | MAG | 2008-03-13 12:00:00 | 2008-06-17 12:00:00 | AE31-769:0701 |
| ch0033a | Magadino-Cadenazzo | MAG | 2008-07-03 13:00:00 | 2008-08-28 10:00:00 | AE31-769:0701 |
| ch0033a | Magadino-Cadenazzo | MAG | 2008-08-28 11:00:00 | 2008-12-05 11:00:00 | AE31-719:0605 |
| ch0033a | Magadino-Cadenazzo | MAG | 2008-12-18 11:00:00 | 2010-03-25 12:00:00 | AE31-719:0605 |
| ch0033a | Magadino-Cadenazzo | MAG | 2010-03-25 11:00:00 | 2013-06-12 13:00:00 | AE31-769:0701 |
| ch0033a | Magadino-Cadenazzo | MAG | 2013-06-12 13:00:00 | 2013-12-18 13:00:00 | AE31-768:0701 |
| ch0033a | Magadino-Cadenazzo | MAG | 2013-12-18 14:00:00 | 2014-07-17 11:00:00 | AE31-719:0605 |
| ch0033a | Magadino-Cadenazzo | MAG | 2014-06-12 09:00:00 | 2016-01-21 11:00:00 | AE31-718:0605 |
| ch0033a | Magadino-Cadenazzo | MAG | 2016-01-21 10:00:00 | 2016-12-21 10:00:00 | AE31-769:0701 |
| ch0033a | Magadino-Cadenazzo | MAG | 2016-12-21 10:00:00 | 2017-05-24 11:00:00 | AE33-S00-00060 |
| ch0033a | Magadino-Cadenazzo | MAG | 2017-05-24 10:00:00 | | AE33-S04-00429 |
| ch2001e | San Vittore | SVI | 2013-10-04 00:00:00 | | AE33-S02-00133 |
| ch0005r | Rigi-Seebodenalp | RIG | 2013-01-22 14:00:00 | 2013-05-30 13:00:00 | AE33-S00-00049 |
| ch0005r | Rigi-Seebodenalp | RIG | 2013-05-30 10:00:00 | 2015-11-12 12:00:00 | AE33-S00-00060 |
| ch0005r | Rigi-Seebodenalp | RIG | 2015-11-12 11:00:00 | | AE33-S02-00136 |





et al., 2003; Drinovec et al., 2015). Generally, the observations were stored as hourly means, but for the data which was at higher resolution (10 and 30 minute means), observations were aggregated to create a consistently hourly time series which was used for analysis. Overall, the analysis covered a period between March 2008 to the end of 2018, but the start date of aethalometer operation varied among the different monitoring sites (Table 1; Table 2).

The method employed to transform absorption observations to EBC requires elemental carbon (EC) concentrations. EC was determined by the standard EN16909 thermal optical transmission (TOT) method using the EUSAAR2 temperature protocol (European Committee for Standardization (CEN), 2017). Unlike the continuous observations which the aethalometers provided, EC concentrations were available as daily samples and ranged from being sampled between every fourth to twelfth day. When using aethalometer and EC data together, the aethalometer observations were aggregated (as arithmetic means) to daily

resolution to ensure the observations spanned the same time period and duration. The EC samples were also collected with $PM_{2.5}$ inlets. $PM_{2.5}$ observations for the monitoring sites were required to calculate the $EBC/PM_{2.5}$ ratios. These data were accessed with the **saqgetr** R package which gives access to the European Commission's AirBase and Air Quality e-Reporting (AQER) repositories in a convenient way (European Environment Agency, 2014, 2019; Grange, 2019b). Only validated data were kept for analysis.

**2.2    Source apportionment**

The absorption measures from the aethalometers were apportioned into two components: the traffic fraction ($EBC_{TR}$) and the woodburning fraction ($EBC_{WB}$) with the aethalometer model (Sandradewi et al., 2008). The aethalometer model is based on the principle that EBC emitted from woodburning activities has an enhanced absorption in the UV range relative to EBC sourced from the combustion of fossil fuels. A critical component of applying the aethalometer model to produce valid outputs

is the selection of the Ångström exponents (which are the slopes of exponential regression models), usually denoted by $\alpha$ (Harrison et al., 2013). Fortunately, in Switzerland, the Ångström exponents for woodburning and traffic sources have been determined robustly using $^{14}$C observations of the EC fraction across Switzerland between 2005 and 2012 (Zotter et al., 2017). Here we use 0.9 and 1.68 for $\alpha_{TR}$ and $\alpha_{WB}$ respectively, as reported by Zotter et al. (2017).

     Mass absorption cross section (MAC) coefficients are required to transform optical absorption observations to EBC mass

(Magee Scientific, 2016). The instrument manufacturers use fixed MAC values on-board to calculate BC mass (for example the A33's factory MAC values are displayed in Table A1), but any given MAC value is a function PM size and morphology which will change depending on the PM source (Zotter et al., 2017). For the trend analysis reported here, the MAC values for absorption at the 950 nm wavelength (used for the IR input to the aethalometer model) were calculated with rolling simple least squares linear regression models between absorption and the mass of EC in the $PM_{2.5}$ fraction. A least squares estimator

was used because EC was determined with a reference method, and is accepted without uncertainty. The window was 180 days (with centre alignment), and this logic was implemented with the **zoo** R package (Zeileis and Grothendieck, 2005).

     Once the Ångström exponents were chosen and the MAC values calculated, the aethalometer model was applied to the absorption observations at hourly resolution and the R function used for this transformation is available (Grange, 2019a). Based on the absorption at 470 and 950 nm (UV and IR spectra respectively), this resulted in three EBC mass variables for





each valid absorption observation: $EBC_{WB}$, $EBC_{TR}$, and $EBC_{TOT}$. A flow diagram which represents the data processing steps can be found in Figure A2.

## 2.3 Trend tests

Formal trend tests were performed on the EBC components with the Theil-Sen slope estimator, a robust, non-parametric
estimator provided by the **openair** R package (Carslaw and Ropkins, 2012). Before the trend was tested, the observations were aggregated to monthly resolution and deseasonalised with loess models to extract the trend component (R Core Team, 2019). Autocorrelation was taken into account resulting in conservative slope estimations and all trend tests were conducted at the 0.05 significance level.

## 3 Results and discussion

### 3.1 Model coefficients evaluation

### 3.1.1 Mass absorption cross section (MAC) coefficients

The MAC values which were empirically derived using least squares regression models (using EC and absorption at 950 nm) with a 180-day rolling window showed substantial variability (Fig. 3). This suggests that the use of fixed or static MAC values are questionable for trend analysis applications. The sites' MAC values at 950 nm generally had a range of $10\,\mathrm{m^2\,g^{-1}}$
during the analysis period despite the irregular cycling of different instruments among the sites (Table 2). For perspective, compare this range with the factory MAC values displayed in Table A1. A seasonal pattern was present in the MAC values where summertime absorption measures were higher than those in wintertime which represent a change of average aerosol characteristics during the year which is to be expected. The range of the seasonality was generally less than $3\,\mathrm{m^2\,g^{-1}}$.

    The MAC values for Rigi-Seebodenalp were atypical and showed a substantial decrease between 2013 and 2016 (Fig. 3).
During this period, the same aethalometer was operating at the monitoring location (Table 2; Fig. A1). The MAC value is an empirical coefficient which represents aerosol composition and it is very unlikely that Rigi-Seebodenalp's aerosol characteristics altered dramatically between 2013 and 2016, therefore the observed decrease in the MAC value was almost certainly an instrument artefact, perhaps a slow degradation in sensitivity. Indeed, the decrease in the MAC value stopped when the instrument was replaced in late-2015 and remained low for the rest of the monitoring period (Fig. A1). This instrument-derived
feature was only uncovered due to the presence of collocated EC observations, but due to a lack of additional data, we do not wish to speculate further on what was the cause for this decline in the MAC coefficient.

    The rolling least squares regression procedure presented here ensures that the changes in MAC over time is compensated for before EBC mass is calculated with the aethalometer model. The changes or drift in the MAC values, especially for Rigi-Seebodenalp, are not propagated into the EBC masses using this method. Therefore, this approach is recommended over using
fixed MAC values if EC time series are available and trend analysis is to be undertaken with several years of aethalometer observations.



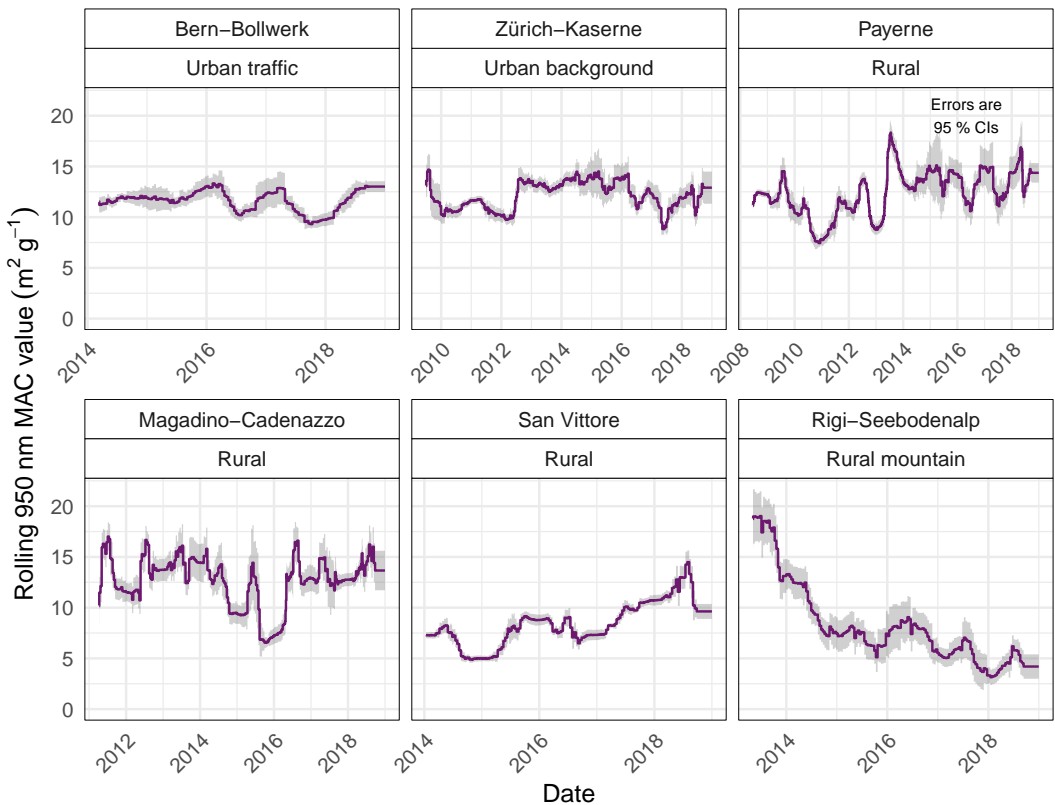

**Figure 3.** Mass absorption cross section (MAC) coefficients for 950 nm for different monitoring sites as calculated by rolling simple linear regression models with windows of 180 days (alignment of the window was the centre of the period).

### 3.1.2 Ångström exponents ($\alpha$)

A critical input for the aethalometer model is the $\alpha$ values used for the traffic and woodburning sourced EBC (Harrison et al., 2013). The $\alpha$ values used here were previously derived from comparing $^{14}$C and EC observations (Section 2.2; Zotter et al. (2017)), but to validate if the values were sensible for the sites analysed, $\alpha$ was calculated for each hourly absorption observation and their distributions investigated.

The distributions of the calculated $\alpha$ values were consistent with those reported by Zotter et al. (2017) which were 0.9 and 1.68 for $\alpha_{\mathrm{TR}}$ and $\alpha_{\mathrm{WB}}$ respectively. Figure 4 shows the $\alpha$ distributions for three of the monitoring sites along a continuum of increasing woodburning activity (from left to right). The distributions for $\alpha$ at each of the monitoring sites peaked at 1.2, the value used for $\alpha_{\mathrm{TR}}$ in the aethalometer model. However, as the monitoring sites' were exposed to a progressively higher woodburning PM load, the distribution become positively skewed. Magadino-Cadenazzo for example, a site which is exposed to wood smoke, has a tail which extends beyond 1.68 while the urban traffic Bern-Bollwerk site's $\alpha$ distribution contains only a





small component at, and beyond, the $\alpha_{\mathrm{WB}}$ value of 1.68. Therefore, the features of the $\alpha$ distributions can be rather informative for diagnostic purposes.

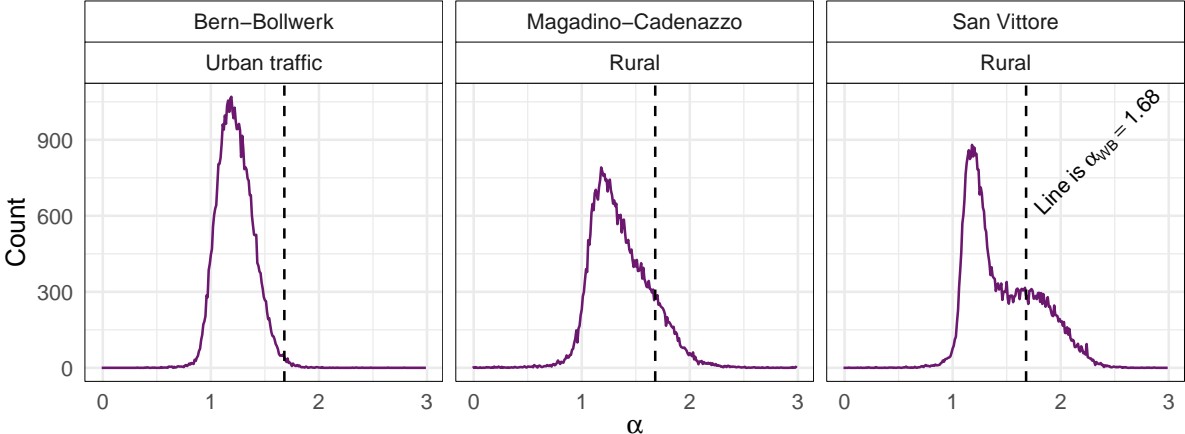

**Figure 4.** Counts of binned Ångström exponents ($\alpha$) for three equivalent black carbon (EBC) monitoring sites' hourly absorption observations in Switzerland between 2014 and 2018.

San Vittore's calculated $\alpha$ were unique because they displayed a bimodal distribution which was not present at the other monitoring sites (Fig. 4). The presence of the bimodal distribution indicates that San Vittore is exposed to an additional source

that the other monitoring sites are not. Although this extra BC source is unknown, we speculate that it is very likely to be freshly emitted wood smoke which has been emitted nearby the monitoring location and rapidly transported to the monitoring site before chemical processing has had the opportunity to act on the aerosol. The importance of this feature is discussed further in Section 3.3. An additional check on the observations was performed in the form of a three-factor receptor model with the multilinear engine (ME-2) as implemented by the EPA PMF tool (Norris et al., 2014; Brown et al., 2015). However, in this case

the factor analysis was unable to resolve the three sources and did not offer any additional contribution to the data analysis.

## 3.2   Diurnal cycles

Diurnal plots were used to validate if the two components' behaviour were consistent with what is expected of their emission source behaviour after the EBC fractions were calculated with the aethalometer model and the model's coefficients evaluated (Fig. 5). With the exception of Rigi-Seebodenalp, an elevated rural mountain site, both the traffic and woodburning components'

diurnal cycles could be readily understood in terms of source activity influencing the monitoring sites. This gives support to the plausibility of the aethalometer model's source apportionment ability.

Bern-Bollwerk, an urban traffic site displayed a clear morning rush hour peak for $\mathrm{EBC_{TR}}$ with a decline in concentrations in the late morning, followed by a smaller increase in the afternoon/evening rush hour (Fig. 5). This traffic-forced pattern was also present at Payerne and Zürich-Kaserne, rural and urban background sites, respectively, but at lower concentrations. Magadino-

Cadenazzo also displayed a clear twin-peak $\mathrm{EBC_{TR}}$ diurnal cycle. Here however, the evening peak had approximately the same



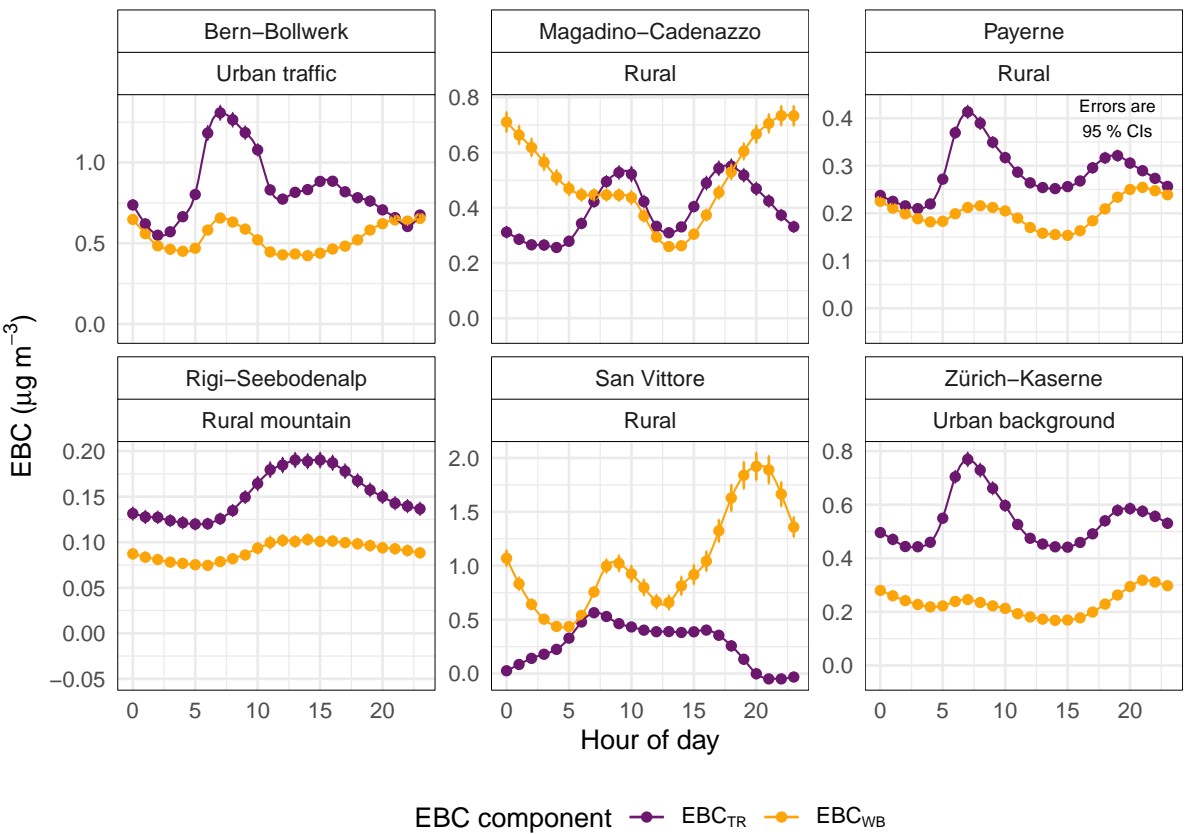

**Figure 5.** Mean hourly equivalent black carbon (EBC) components for the six monitoring sites. Note the different scales on the $y$-axes.

concentration as the morning peak which was not observed at the other monitoring locations. The strong evening peak was only observed at winter at Magadino-Cadenazzo, This feature was most likely driven by very stable and stagnant atmospheric conditions in the morning and evening because this monitoring site (and surrounding population centres), is located in a deep valley where the day length is short at these times of the year due to confining terrain. This is confirmed by mean bivariate

5    polar plots for Magadino-Cadenazzo where elevated $EBC_{WB}$ concentrations were found when wind speeds were very low ($\leq 2\,\mathrm{m\,s^{-1}}$) while $EBC_{TR}$ sources were identified for a diverse range of wind speeds and directions (Fig. 6).

For most monitoring sites, the $EBC_{WB}$ components displayed elevated concentrations in the late evening with a decline in the early morning, consistent with domestic heating demands (Grange et al., 2013). In most cases, this evening peak was followed by decrease and a lesser morning increase in $EBC_{WB}$, likely due to the reignition of woodburning appliances in the

10    morning. However, at the Bern-Bollwerk urban traffic site, the morning $EBC_{WB}$ peak was reminiscent of the $EBC_{TR}$ peak, indicating a slight contamination of $EBC_{WB}$ by $EBC_{TR}$ which the aethalometer model was unable to separate. Magadino-





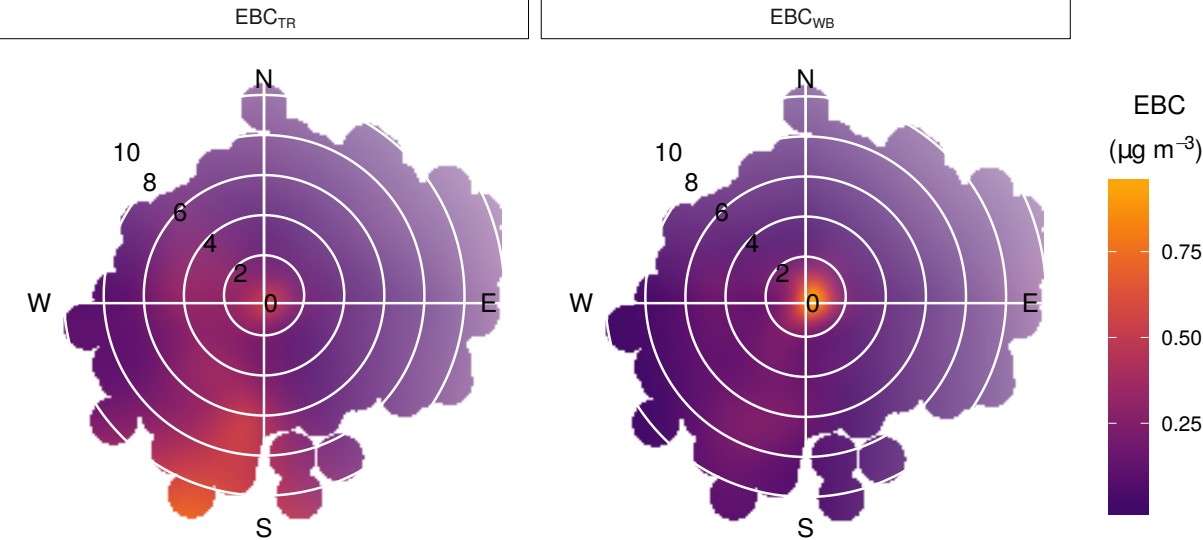

**Figure 6.** Mean polar plots of equivalent black carbon (EBC) components for Magadino-Cadenazzo between 2011 and 2018.

Cadenazzo, the rural monitoring location which is exposed to a high load of woodburning PM, the diurnal cycle was very clear and strong.

Rigi-Seebodenalp's EBC diurnal cycles were different than the other monitoring locations (Fig. 5). This monitoring site is located at an elevation of 1031 metres, is isolated from any significant local emissions, and is intermittently in the boundary

5  layer and therefore, at times is not influenced by surface source activities (Grange et al., 2018). These site attributes resulted in EBC diurnal cycles being driven primarily by boundary layer evolution rather than local source strength and activities. When the convective boundary layer grew in vertical extent and exceeded the site's elevation, EBC was mixed to the site, but when the stratified nocturnal boundary layer decoupled the site from surface-based emissions, concentrations remained low (Fig. 5). This feature was clearer for $EBC_{TR}$ than $EBC_{WB}$.

10  **3.3  Aethalometer model failure**

San Vittore is a small settlement located in the Mesolcina Valley in the south of Switzerland (Fig. 2), an area which experiences high concentrations of PM and BC from residential woodburning. The diurnal plots for San Vittore demonstrate, on average, negative mass contributions for $EBC_{TR}$ when $EBC_{WB}$ concentrations were high in the evening (averages shown in Fig. 5). This is an implausible situation and indicates the failure of the aethalometer model to correctly apportion the traffic and

15  woodburning EBC sources. When the time series of the $EBC_{TR}$ and $EBC_{WB}$ are plotted, the negative contributions are very clear during the colder months, November to December (Fig. 7).

These negative contributions can be managed to some extent by altering the woodburning Ångström exponent ($\alpha_{WB}$), but only with justification. In the case of San Vittore, the $\alpha_{WB}$ was increased to $\approx 2.0$ (a very high value) in an attempt to eradicate



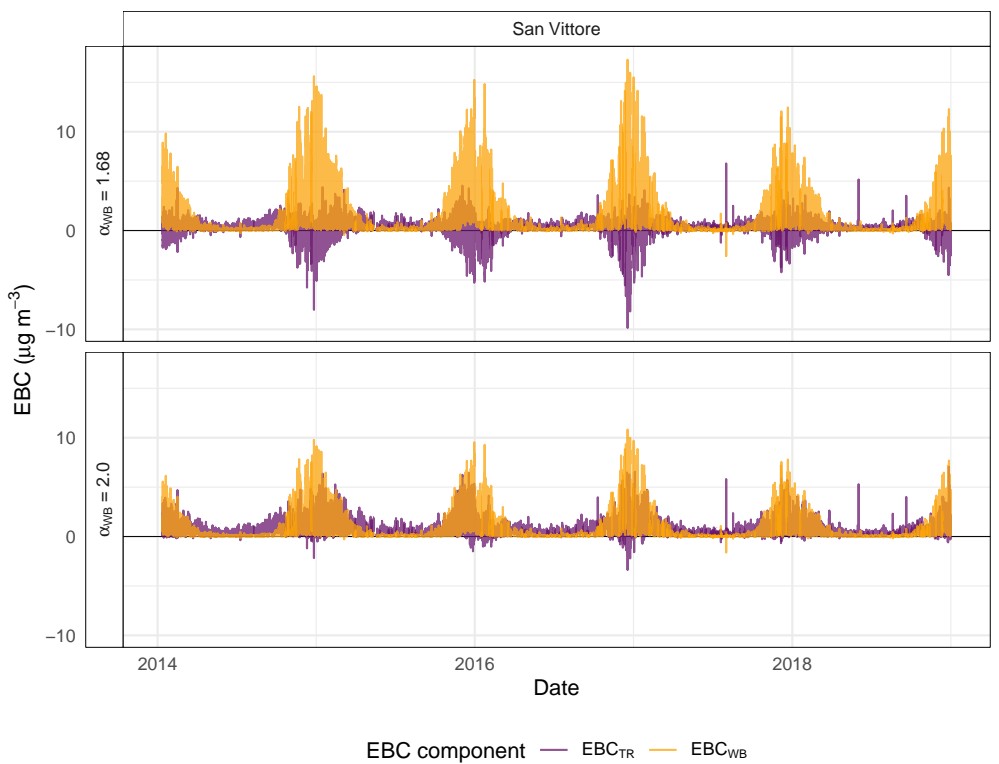

**Figure 7.** $EBC_{TR}$ and $EBC_{WB}$ hourly time series between 2014 and 2018 at San Vittore using two values for $\alpha_{WB}$. The $\alpha_{WB}$ of 1.68 was used for the analysis but showed negative contributions of $EBC_{TR}$ when $EBC_{WB}$ concentrations were high and using an $\alpha_{WB}$ of 2.0 showed implausible $EBC_{TR}$ concentrations at this site.

the negative $EBC_{TR}$ mass contributions (Fig. 7). This solution only partially resolves the negative $EBC_{TR}$ concentrations, but also results in the $EBC_{TR}$ being estimated at implausibly high concentrations for this location. This is most evident during the cooler periods where $EBC_{WB}$ concentrations are high due to a heavy burden of wood smoke, but $EBC_{TR}$ concentrations are also high and based on San Vittore's characteristics, we concluded that the aethalometer model is unable to separate the two

5 EBC components in this time series.

The San Vittore example represents a very clear example of when the aethalometer model fails and the results cannot be used further for data analysis. The aethalometer model fails in this example because the monitoring site is exposed to freshly emitted wood smoke from the near-by residential properties. Woodburning aerosol which has not been exposed to any or very little atmospheric ageing has different special properties than those exposed to atmospheric processing (Jimenez et al., 2009)

10 (which was indicated in Fig. 4) and is the explanation for the aethalometer model failure. The aethalometer model's two-source approach is most likely insufficient for the freshly emitted wood smoke which may need to be treated as a third distinct source. A receptor modelling approach (such as PMF) may be a technique which could resolve these three sources, but an investigation





of this was outside of the scope of this study. Observations from the San Vittore site were excluded from any further analysis but have been presented to demonstrate that although the aethalometer model can be a useful technique, it is pragmatic and is not appropriate in all cases.

### 3.4 EBC dependence on air temperature

BC emissions from traffic and woodburning sources are not only different in respect to their timing as discussed in Section 3.2, but also by their dependence on ambient air temperature. Although vehicle emissions of PM are known to be temperature dependent (Jamriska et al., 2008; Weilenmann et al., 2009), this effect can be expected to be less pronounced when compared to residential heating emissions which have a quite distinct "heating threshold temperature" where emissions will increase as the air temperature decreases further to meet an increasing space heating energy demand. Additionally, in warmer periods,
there is very little or no emissions of BC from residential heating.

These different source patterns were clearly demonstrated for the Swiss EBC monitoring sites with Magadino-Cadenazzo $EBC_{WB}$ concentrations being negative correlated with air temperature, and with increases in mean $EBC_{WB}$ concentrations once air temperatures were below $12\,^{\circ}C$ (Fig. 8). Temperatures of $17\,^{\circ}C$ and above resulted in $EBC_{WB}$ being less than $0.15\,\mu g\,m^{-3}$ at the same location. Payerne's EBC components showed similar patterns to Magadino-Cadenazzo while the
sites which were influenced more by traffic emissions, Bern-Bollwerk and Zürich-Kaserne still demonstrated a greater ambient temperature dependence for $EBC_{WB}$ than $EBC_{TR}$ despite $EBC_{TR}$ concentrations being higher for most temperatures.

Due to Rigi-Seebodenalp's elevated location, the patterns observed are different than at the other monitoring sites, in particular as temperature increased so did $EBC_{TR}$ concentrations (Fig. 8). This was a result of mixing of traffic sourced pollutants to the monitoring location and shows that Rigi-Seebodenalp is influenced more by traffic emissions than those originating from
woodburning activities which was also suggested by Fig. 5.

### 3.5 Trend analysis

After the aethalometer model split EBC into the two components, $EBC_{TR}$ was found to be significantly decreasing across all monitoring sites in Switzerland between 2008 and 2018, with the exception of the rural mountain monitoring site, Rigi-Seebodenalp (Fig. 9). Bern-Bollwerk, an urban traffic site was the most polluted EBC monitoring site (Table A2) and saw the
greatest reduction in $EBC_{TR}$ with a decrease of $-0.13\,\mu g\,m^{-3}$ year$^{-1}$. Magadino-Cadenazzo, Payerne, and Zürich-Kaserne demonstrated modest significantly decreasing $EBC_{TR}$ trends ranging from $-0.028$ to $-0.059\,\mu g\,m^{-3}$ year$^{-1}$ (Fig. 9).

These $EBC_{TR}$ trends can be interpreted as successful management of BC emissions from vehicular internal combustion engines. There were no obvious breakpoints in the trend components, but the widespread installation of diesel particle filters (DPF) for the (late) Euro 4 and 5 emission standards (beginning in 2000) and the elimination of older diesel-fuelled vehicles
are most likely the main drivers for the reductions in concentrations of $EBC_{TR}$ (Gill et al., 2011; Reşitoğlu et al., 2015). The efficacy of DPFs for trapping soot post-combustion chamber has offset the high proportion of diesel passenger vehicles in the European fleet which peaked in 2010 at over 50 % (Schiermeier, 2015; International Council on Clean Transportation Europe, 2015; Zachariadis, 2016; ACEA, 2018). However, the diesel market share for passenger vehicles sharply declined in the wake

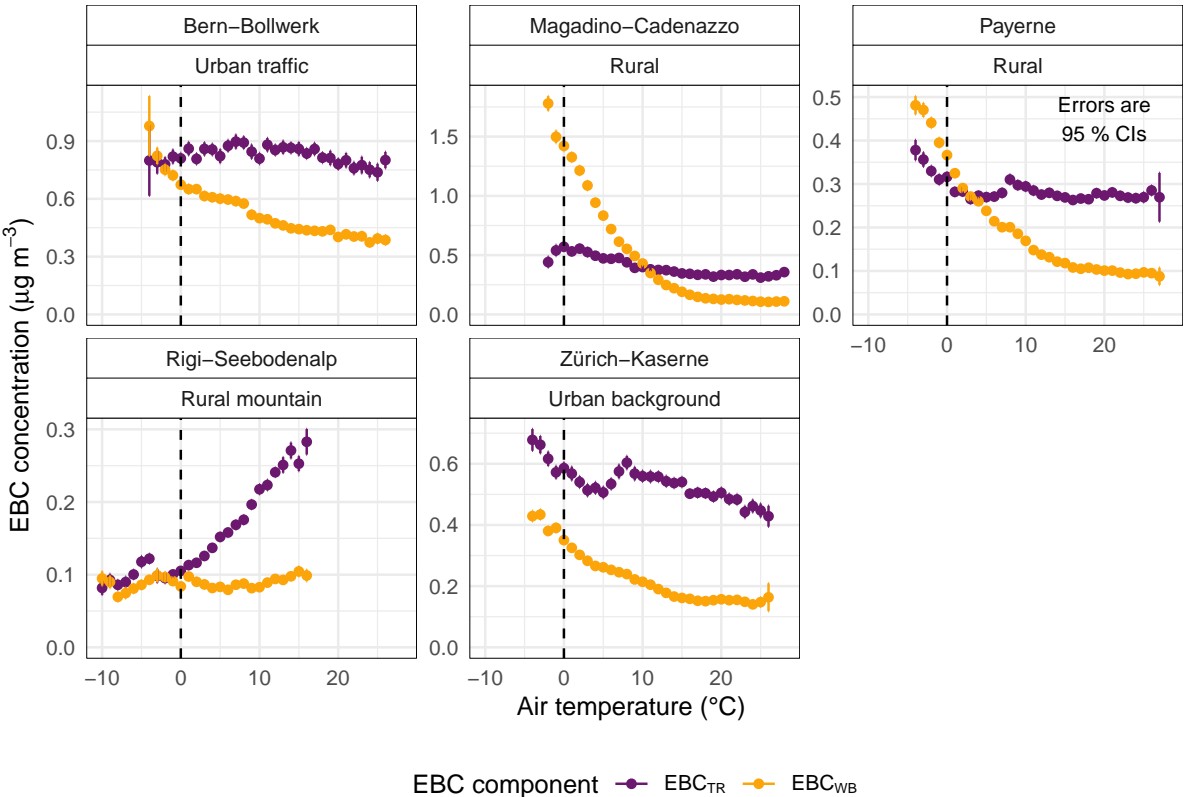

**Figure 8.** $EBC_{TR}$ and $EBC_{WB}$ dependence on air temperature for five equivalent black carbon (EBC) monitoring sites in Switzerland.

of the Volkswagen emission scandal ("dieselgate") after 2015, and this market change may also be a contributory factor in the $EBC_{TR}$ trends observed after about 2016.

    $EBC_{WB}$ trends were different than those observed for $EBC_{TR}$, even when comparing the same monitoring sites (Fig. 9). The urban monitoring locations, Bern-Bollwerk and Zürich-Kaserne demonstrated significantly decreasing $EBC_{WB}$ trends

5  but at a smaller magnitude when compared to the $EBC_{TR}$ components at the same sites. Payerne's $EBC_{WB}$ concentrations were also found to be decreasing significantly, but at a minute rate (-0.005 $\mu g\,m^{-3}$ year$^{-1}$). Critically, the monitoring site which experiences the greatest woodburning PM load, Magadino-Cadenazzo south of the Alps (Fig. 2), showed no significant trend in the $EBC_{WB}$ component (Fig. 9). This is in contrast to the traffic-sourced BC and indicated that efforts to reduce PM emissions from woodburning activities have not been successful in communities south of the Alps. The same conclusion was

10  found when $PM_{10}$ observations were previously analysed at the same monitoring site (Grange et al., 2018).

    Figure 5 and 8 suggest that Rigi-Seebodenalp's isolation from localised BC sources made the site distinct from the other EBC monitoring sites in Switzerland. The trends for Rigi-Seebodenalp were also distinct and showed no significant trend in both the $EBC_{TR}$ and $EBC_{WB}$ components (Fig. 9).





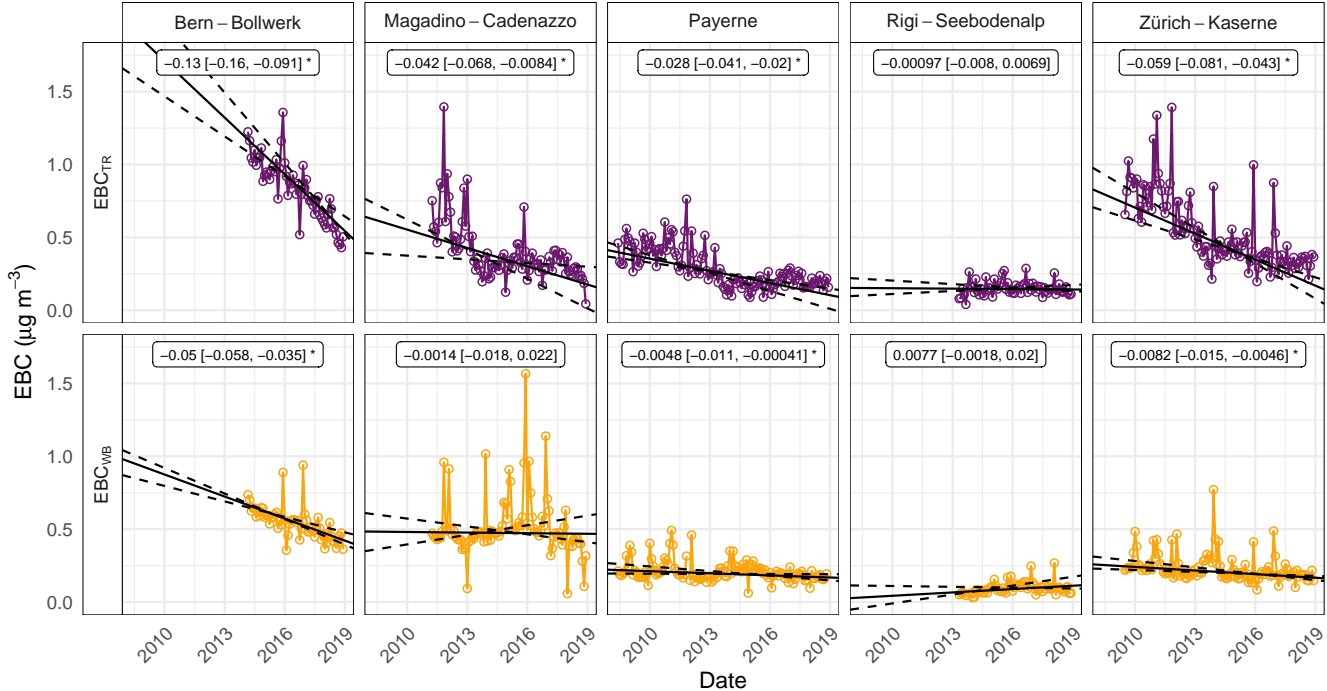

**Figure 9.** Deseasonalised trends of $EBC_{TR}$ and $EBC_{WB}$ at five EBC monitoring sites in Switzerland between 2008 and 2018. Asterisks (*) in the panels indicate if the trend was statistically significant and dashed lines show the trend tests' 95 % confidence interval.

### 3.6 EBC/PM$_{2.5}$ ratios

BC is almost exclusively an anthropogenic pollutant with generally minor and intermittent contributions from "naturally" caused combustion processes. It is therefore useful to explore what effect a complete eradication of the BC emission sources would have on air quality. The seasonal EBC/PM$_{2.5}$ ratios for the five Swiss EBC monitoring sites are shown in Fig. 10

5 and shows that the ratios were variable among the monitoring sites and seasons. Bern-Bollwerk, the urban traffic site had the highest EBC/PM$_{2.5}$ ratio while Payerne and Rigi-Seebodenalp had the lowest. Again, Fig. 10 gave plausibility to the aethalometer model's source apportionment abilities because features such as $EBC_{WB}$ having very low contributions in the summer for most sites, even for the biomass burning dominated Magadino-Cadenazzo location. However, as suggested by Fig. 5, Bern-Bollwerk's $EBC_{WB}$ contribution to PM$_{2.5}$ were most likely too high, indicating that that the aethalometer model

10 was been unable to fully separate the two $EBC_{TR}$ and $EBC_{WB}$ fractions completely for this particular monitoring site.

On first inspection, the $EBC_{TOT}$ contribution to PM$_{2.5}$ mass seems modest, ranging from 6–13 % depending on site and season. Compare these values to the largest European urban areas, Paris and London, where their BC/PM$_{2.5}$ ratios have been reported (at single heavily trafficked locations) as 43 ± 20 % and ≈ 50 % respectively (Ruellan and Cachier, 2001; Grange et al., 2016). If BC was perfectly controlled at the source of emission, Switzerland could achieve a 6–13 % reduction of PM$_{2.5}$





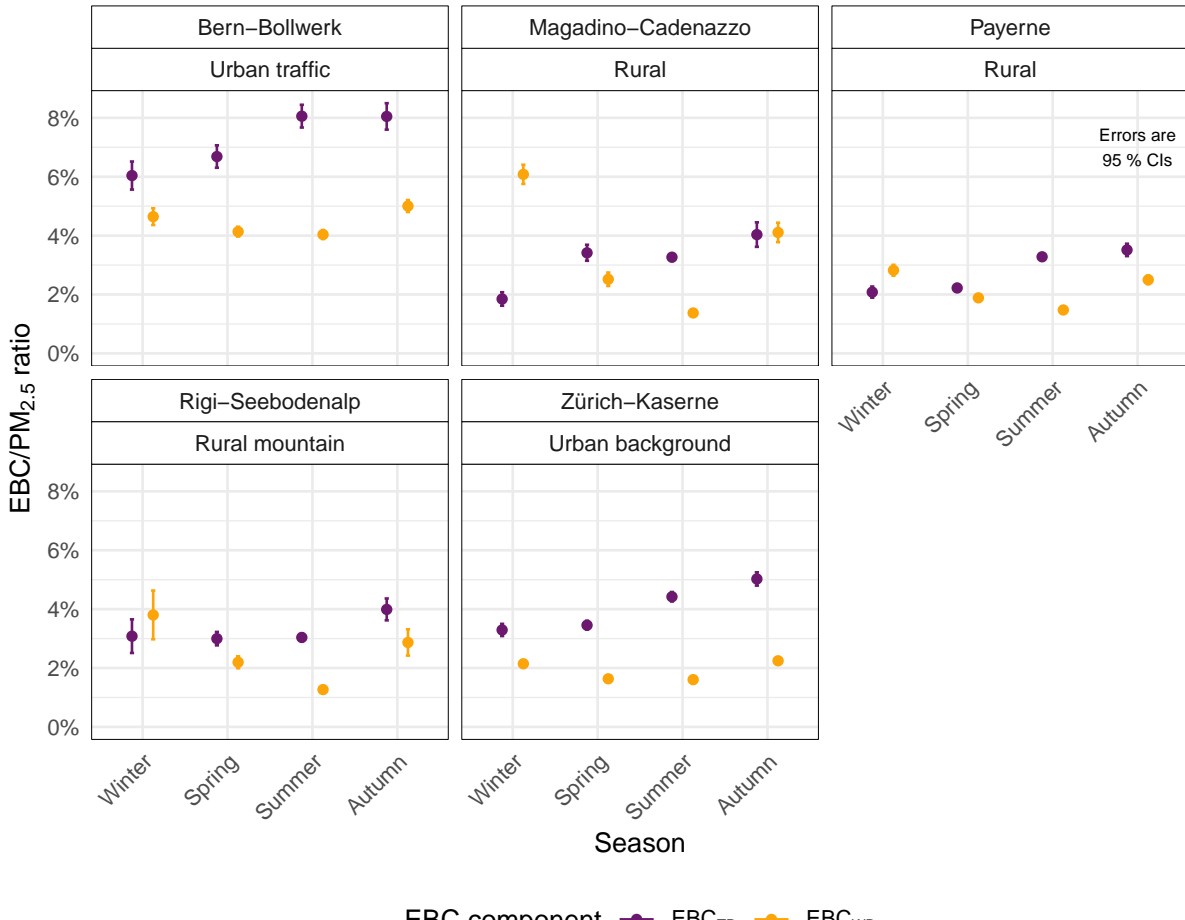

**Figure 10.** Seasonal equivalent black carbon (EBC)/PM$_{2.5}$ ratios for five monitoring sites in Switzerland between 2014 and 2018.

concentrations to aid with the compliance with the annual and daily limits set for PM$_{2.5}$, most notably Switzerland's tight annual 10 $\mu$g m$^{-3}$ year$^{-1}$ limit introduced in 2018.

The discussion above is focused on the contribution of EBC to PM. In addition, BC should be to reduced to the lowest possible level because of its classification as a Group 1 carcinogen by the International Agency for Research on Cancer (IARC;

5    entered as soot) (International Agency for Research on Cancer, 2019). The reduction of soot emissions also has co-benefits for reducing climate warming effects and this can also be used for motivation for the implementation of soot control at the source.





## 4   Conclusions

Using aethalometers for EBC monitoring allows for the application of the aethalometer model to split EBC into $EBC_{TR}$ and $EBC_{WB}$ components. The aethalometer model is a useful and pragmatic data processing technique, but it requires evaluation before using the model's outputs for further data analysis activities. Based on the results presented using Swiss aethalometer observations from six sites between 2008 and 2018, these recommendations can be offered to other data users:

– Values used for the mass absorption cross section (MAC) coefficients and the Ångström exponents ($\alpha$) should be checked with the observational record which is being analysed to ensure they are plausible for use with the particular data set. EC observations are however required for the calculation of MACs and therefore, the expansion of monitoring networks to include regular EC samples is recommended for better exploitation of aethalometer absorption data.

– Evaluate the calculated $EBC_{TR}$ and $EBC_{WB}$ by investigating the presence of negative mass contributions, their diurnal cycles, and ambient temperature dependence to ensure these features are consistent with what is known about the sites' PM load and relevant physical and chemical atmospheric processes.

– If available, put the $EBC_{TR}$ and $EBC_{WB}$ in context by using PM mass concentrations.

These recommendations will help those who are interesting in applying the aethalometer model which has a rather low barrier to entry.

Despite the failure of the aethalometer model to produce value outputs for one of the six monitoring sites analysed, $EBC_{TR}$ and $EBC_{WB}$ were successfully analysed separately at five locations, and different trends were observed for the two EBC components. $EBC_{TR}$ concentrations significantly decreased across all monitoring sites between 2008 and 2018 with the exception of an isolated rural mountain monitoring location at a maximum rate of -0.13 $\mu g\,m^{-3}$ year$^{-1}$. It is likely that the $EBC_{TR}$ trends were primarily driven by the widespread installation of effective DPF throughout the in-service heavy- and light-duty vehicle fleet. Trends of $EBC_{WB}$ were more variable and despite some significantly deceasing trends of some monitoring sites, a site known to be heavily burdened by wood smoke showed no signifiant trend over the 2008 and 2018 monitoring period. This is evidence of ineffective management of reducing BC and PM emissions from domestic wood burning which has been noted in the past. The $EBC/PM_{2.5}$ ratios for the five monitoring sites showed location and seasonal variability, and the EBC contribution to $PM_{2.5}$ was between 6 and 13 %. This $EBC/PM_{2.5}$ ratio is low when compared to other heavily trafficked sites in Paris and London, but is important to know for an air quality management perspective. The aethalometer model is a pragmatic and useful data processing technique, but full evaluation is needed to ensure the results are suitable for further data analysis.

*Code and data availability.*   The data sources used in this work are described and referenced in the text. Observations for the Bern-Bollwerk and San Vitore monitoring sites (which are not currently public accessible) are available from the authors on reasonable request.



*Author contributions.* SKG and CH developed the research questions and conducted the data analysis. HP and AF collected and supplied additional data. SKG and CH prepared the manuscript with input from LE.

*Competing interests.* The authors declare no competing interest.

*Acknowledgements.* This work was financially supported by the Swiss Federal Office for the Environment (FOEN). SKG is also supported
5   by the Natural Environment Research Council (NERC) while holding associate status at the University of York. The authors thank the COST
Action CA16109 COLOSSAL (Chemical On-Line cOmpoSition and Source Apportionment of fine aerosoL) for motivating this research.





**Table A1.** Factory mass absorption cross section (MAC) coefficients for the AE33 aethalometer (Magee Scientific, 2016).

| Wavelength (nm) | MAC value ($\mathrm{m^2\,g^{-1}}$) | Notes |
|---:|---:|---|
| 370 | 18.47 | |
| 470 | 14.54 | |
| 520 | 13.14 | |
| 590 | 11.58 | |
| 660 | 10.35 | |
| 880 | 7.77 | The definition of black carbon (BC) |
| 950 | 7.19 | |

**Table A2.** Summary statistics for daily equivalent black carbon (EBC) at five monitoring sites in Switzerland between 2008 and 2018.

| Site name | Site type | $\mathrm{EBC_{TOT}}$ ($\mathrm{\mu g\,m^{-3}}$) | $\mathrm{EBC_{TR}}$ ($\mathrm{\mu g\,m^{-3}}$) | $\mathrm{EBC_{WB}}$ ($\mathrm{\mu g\,m^{-3}}$) |
|---|---|---|---|---|
| Bern-Bollwerk | Urban traffic | 1.37 (95% CI [1.34, 1.39]) | 0.83 (95% CI [0.82, 0.85]) | 0.53 (95% CI [0.52, 0.55]) |
| Zürich-Kaserne | Urban background | 0.77 (95% CI [0.76, 0.79]) | 0.54 (95% CI [0.53, 0.55]) | 0.23 (95% CI [0.23, 0.24]) |
| Payerne | Rural | 0.49 (95% CI [0.48, 0.50]) | 0.29 (95% CI [0.28, 0.29]) | 0.20 (95% CI [0.20, 0.21]) |
| Magadino-Cadenazzo | Rural | 0.90 (95% CI [0.87, 0.93]) | 0.39 (95% CI [0.38, 0.41]) | 0.50 (95% CI [0.48, 0.52]) |
| Rigi-Seebodenalp | Rural mountain | 0.24 (95% CI [0.23, 0.25]) | 0.15 (95% CI [0.15, 0.16]) | 0.09 (95% CI [0.09, 0.10]) |

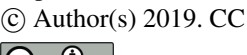

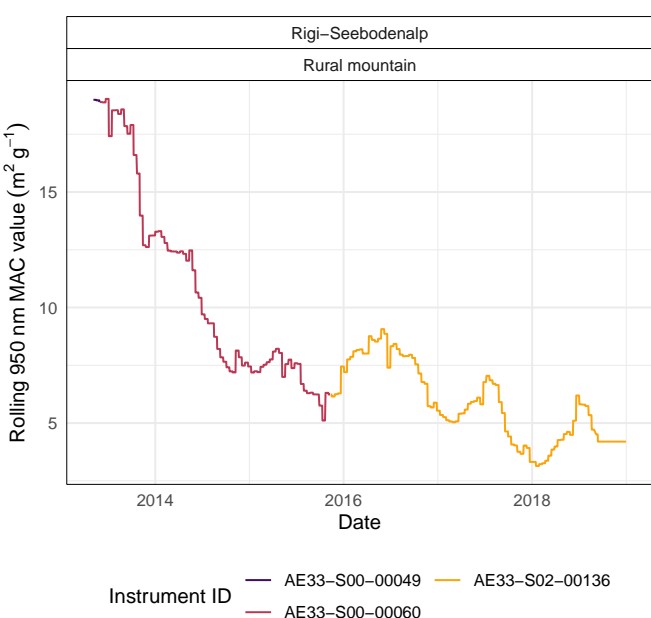

**Figure A1.** Mass absorption cross section/mass absorption coefficients (MAC) values for 950 nm at Rigi-Seebodenalp and coded by the aethalometers which were operating at the monitoring site.



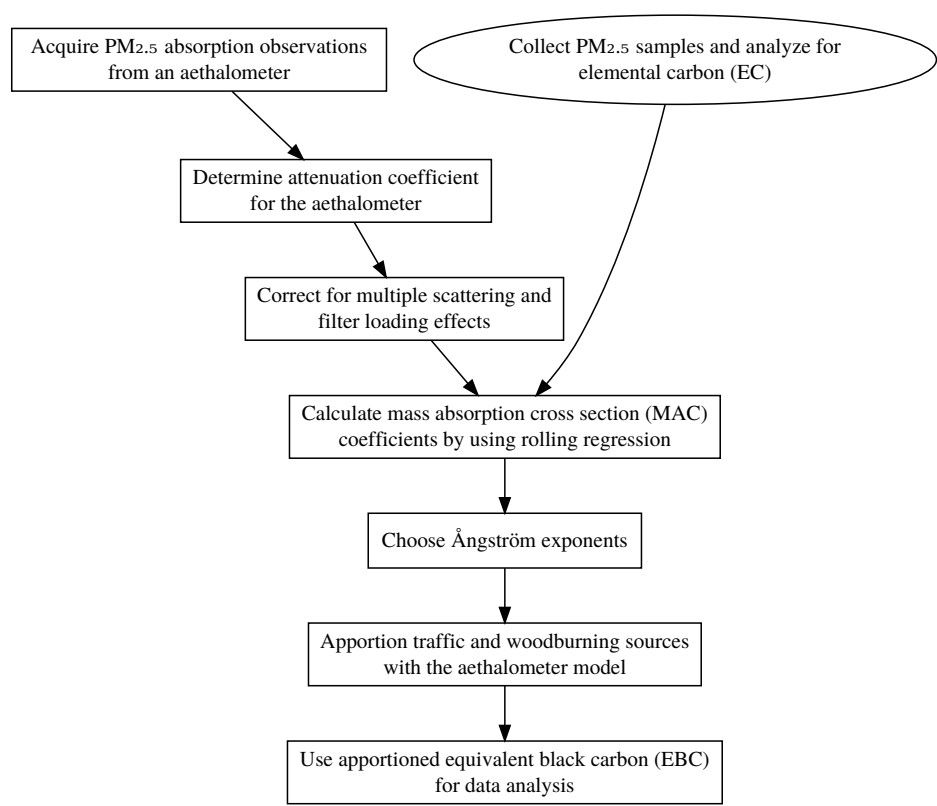

**Figure A2.** A flow chart of the data processing steps required to apply the aethalometer model.





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
