# Peer review of "Evaluation of equivalent black carbon (EBC) source apportionment using observations from Switzerland between 2008 and 2018"

_Atmospheric Measurement Techniques, 2019_

## Referee Comment (RC1) · Anonymous Referee #2 · 4 Dec 2019

This manuscript reports trends in equivalent black carbon (EBC) at several sites in Switzerland. Using aethalometer data, the authors apply a model to apportion EBC to traffic and wood burning sources that requires using Angstrom exponents and the assumed spectral dependence of wood smoke and traffic emissions. The data were interpreted in the context of these sources, including seasonality and diurnal patterns. The authors also report analysis of long-term trends in both the traffic and wood burning fractions. Results corresponding to periods when the aethalometer model failed provided additional feedback for when it is unable to successfully apportion sources, in this case in situations with freshly emitted wood smoke. The paper is clearly written and organized and the methods are well described and sound. The results are helpful in understanding the role of emission mitigation strategies for traffic and biomass burning sources. I recommend publication after addressing comments below.

P1, Line 7: Given the issues the model had with what appeared to be fresh smoke, would it be fair to refer to this fraction as aged?

P1, Line 12: This goes back to the previous comment. To clarify, calling the smoke fraction "aged woodburning" would help distinguish these sources.

P1, Line 15: Change "deceases" to "decreases"

P1, Line 16: What does "This" refer to at the beginning of the sentence?

P1, Line 17: If the site is a likely representative location and EBCWB has not decreased, but the other sites have, how is it representative of ineffective controls on wood burning? Are the same management controls applied everywhere?

P2, Line 6: Change "fuelled" to "fueled"

P7, Line 2: Change "consistently" to "consistent"

P7, Line 8: What period was considered a daily sample? Midnight-midnight?

P7, Line 10: How were EC data applied with this frequency since BC and EC would overlap only on certain days? Was the EC sampling schedule the same at all sites?

P7, Line 11: How was PM2.5 measured at all of the sites? What sampling frequency?

Page 7, Line 16: Change "measures" to "data"

Page 8, Line 4: Were any trend analyses performed on the EC data to test whether EC trends generally followed the overall BC trends?

Page 8, Line 16: Where was this seasonal pattern observed? At all sites? A range of 3 m2/g is very large, can the authors comment on the physical reasons why the MAC would vary this much on a seasonal basis? They also vary considerably from site to site. Were EC data examined on a site by site and seasonal basis? Do they show this

much variability?

Page 9, Line 6: Can the authors provide a brief explanation on how Angstrom exponents were calculated? Are they ratio of two wavelengths or a fit of all wavelengths?

Page 10, Line 2, Figure 4 caption. Can the authors add the wavelength range for the Angstrom exponents?

Page 12, Line 4: What about wood burning as local emissions?

Page 12, Line 9: Why would this feature be stronger for traffic than wood burning sources?

Page 14: Line 23: It appears from the figures that a couple of sites do not have data before 2013. Trend analyses performed over this short of period can be misleading when compared to sites with longer periods.

Page 14: Line 26: The Payerne site appears to have a break around 2014, after which the data are relatively flat. A similar pattern may occur at Magadin-Cadenazzo and Zurich sites. Is this an instrument artifact?

Page 15: Line 12-13: But the time period is much shorter and during this period, other sites had flat trends (Payerme and Magadino), so it may not be fully reflective of what is happening over the longer time period.

Page 16, Line 5: Over the same years?

---

## Referee Comment (RC2) · Anonymous Referee #3 · 16 Dec 2019

Grange et al. present an evaluation of source apportionment via the aethalometer model from six Swiss sites. They compare their obtained fossil and wood burning fractions from the model to expected diurnal patterns and ambient temperature. They also compare the aerosol Ångstroem exponents (AAE) they use in the model to the binned AAEs of the full (non-apportioned) time series. Finally, they also show and discuss the trends in fossil/ wood burning fractions. This article is informative the figures are well designed and illustrative. However, there are some important issues regarding the data processing, at least as it is outlined in the text. I also note that the technical aspects of the paper are not as in depth as work by e.g. (Fuller et al., 2014) or (Zotter et al., 2017) who compare their data to external tracers, such that an experienced user

of aethalometer data might learn more from the reported trends and source appor-
tionment than the technical aspects presented. Nevertheless I recommend publication
after the following issues are addressed.

Major issues:

Most data in the paper are from the Swiss National Air Pollution Monitoring Network
(NABEL). I tried to access this site using the link in the references, but found that
the link was no longer active, whereas the authors state that data are available as
described in the text. Furthermore there is no description of data quality control /
assurance. Are all data aggregated to hourly resolution or are some data points re-
moved due to instrumental issues etc.? If points are removed, what is the threshold
at which an hourly data point is still reported (i.e. what number of missing points are
allowed?). The authors should update the information they provide on data availabil-
ity and describe the quality control measures. Payerne and Rigi are EMEP sites and
aethalometer data are publicly available at ebas.nilu.no.

The authors do not seem to have access to time series at the original time resolution
of the instruments since they write 'Generally, the observations were stored as hourly
means, but for the data which was at higher resolution (10 and 30 minute means)',
implying that raw instrument output was not available? Checking these time series
is important for several reasons, e.g. to ensure correct application of the automatic
loading compensation in the AE33 instruments (i.e. there should be no structure in
the absorption or EBC time series related to tape advances), or that no short term
spikes are present in the data which could bias the averaging. Plotting the AAE over
time is also useful. This lack of information weakens the arguments the authors make
regarding the failure of the aethalometer model apparent at the San Vittore site. In the
text they assert that this failure is due to the presence of a third source (fresh wood
smoke). Here it is important to rule out other causes of failure, which might arise due
to errors in the data processing (e.g. incorrect loading compensation). If possible, time
series should be shown in the annex. If not, this weakness should be mentioned in the

text.

Minor comments

Change 'appointment' in the title to 'apportionment'.

Pg 1, ln 16: change 'has be' to 'has been'

Pg 1, ln 17: PM2.5 is undefined

Pg 2, ln 19: Suggest using 'direct legal limits' since in the unlikely event that BC is itself above the limit values for PM2.5 then BC would be above the legal limit. Non-binding provisions for BC limits are also included in the Gothenburg protocol, see also (Shapovalova, 2016).

Pg 5, ln 12: Data from AE31 instruments are included and the authors write 'All absorption observations had been compensated for the filter loading and shadowing effects with the instrument model's respective algorithms before this analysis was undertaken'. However, the output from the AE31 is slightly different to that of the AE33 in that a correction factor for loading is not automatically applied by the instrument and must be done in post processing. Please clarify.

Pg 7, ln 9: The authors write 'When using aethalometer and EC data together, the aethalometer observations were aggregated (as arithmetic means) to daily 10 resolution'. I assume the data were averaged to the start and end times of the filters? In which case it would be better to write something like 'data were window averaged to the start and end times of the filter sampling'.

Pg 7, ln 23: Use 'recommended by' instead of 'reported by'.

Pg 8, ln 12: What was the least squares algorithm used? Given that there is uncertainty in both x and y, orthogonal distance regression should be used.

Pg 8, ln 14: Here it would be better to use 'an average of', rather than a 'range of' since the range is later given as 3 m2/g. I.e. write an average of 10±3 m2/g

Pg 8, ln 18. The report by Anonymous Referee #2 discusses the issue of the large variability in MAC. How much of the variation in MAC is actually due to instrument differences? It is natural to expect some variability due to this. The effect of this will have been somewhat obscured by using the rolling average presented in the paper, so that Fig. A1 likely does not show the full magnitude of the effect. It would be informative to include e.g. seasonal MAC values on an instrument by instrument basis e.g. for each line of Table 2. A version of Fig. 3 highlighting instrument shifts would also be informative.

Pg 8, ln 27: change 'is' to 'are'.

Pg 14, ln 12: change 'negative' to 'negatively'

Pg 22, Figure A2 caption: Please change the caption to reflect that these are the steps used in the paper and not a general requirement, e.g. to 'A flow chart of the data processing steps used to apply the aethalometer model'. Some of the steps could have been done in a different order, and apportionment of absorption coefficients is also possible without conversion to EBC.

References

Fuller, G. W., Tremper, A. H., Baker, T. D., Yttri, K. E., and Butterfield, D. J. A. e.: Contribution of wood burning to PM10 in London, 87, 87-94, 2014.

Shapovalova, D.: The Effectiveness of Current Regulatory Models of Gas Flaring in Light of Black Carbon Emissions Reduction in the Arctic, in: Global Challenges in the Arctic Region: Sovereignty, Environment, and Geopolitical Balance, Routledge, 325-344, 2016.

Zotter, P., Herich, H., Gysel, M., El-Haddad, I., Zhang, Y., Močnik, G., Hüglin, C., Baltensperger, U., Szidat, S., Prévôt, A. S. J. A. c., and physics: Evaluation of the absorption Ångström exponents for traffic and wood burning in the Aethalometer-based source apportionment using radiocarbon measurements of ambient aerosol, 17, 4229-

4249, 2017.

---

## Author Comment (AC1) · 29 Jan 2020

**Author responses to Referee #2's comments of amt-2019-351 (*Evaluation of equivalent black carbon (EBC) source apportionment using observations from Switzerland between 2008 and 2018*)**

January 29, 2020

Stuart K. Grange*, Hanspeter Lötscher, Andrea Fischer, Lukas Emmenegger, and Christoph Hueglin

*stuart.grange@empa.ch

**Response to reviewers**

**Referee #2**

This manuscript reports trends in equivalent black carbon (EBC) at several sites in Switzerland. Using aethalometer data, the authors apply a model to apportion EBC to traffic and wood burning sources that requires using Angstrom exponents and the assumed spectral dependence of wood smoke and traffic emissions. The data were interpreted in the context of these sources, including seasonality and diurnal patterns. The authors also report analysis of long-term trends in both the traffic and wood burning fractions. Results corresponding to periods when the aethalometer model failed provided additional feedback for when it is unable to successfully apportion sources, in this case in situations with freshly emitted wood smoke. The paper is clearly written and organized and the methods are well described and sound. The results are helpful in understanding the role of emission mitigation strategies for traffic and biomass burning sources. I recommend publication after addressing comments below.

Thank you. We have addressed all of the referee's comments below.

1. P1, Line 7: Given the issues the model had with what appeared to be fresh smoke, would it be fair to refer to this fraction as aged?

   This point is addressed below in point 2.

2. P1, Line 12: This goes back to the previous comment. To clarify, calling the smoke fraction "aged woodburning" would help distinguish these sources.

   These two above points can be addressed together. The woodburning fraction which the aethalometer model apportions is considered a mixture of fresh and somewhat aged woodburning sourced BC. In a typical urban area, such as Zürich, the woodburning sourced BC

encountered will be a combination of aged particles as well as freshly emitted emissions. We believe the model failure shown in San Vittore was caused by a much greater proportion of fresh woodburing emissions which has different spectral characteristics than the more commonly encountered mixed woodburning BC. Therefore, calling the woodburning fraction 'aged' does not seem to be truly correct here.

3. P1, Line 15: Change "deceases" to "decreases"

Done.

4. P1, Line 16: What does "This" refer to at the beginning of the sentence?

The sentence now begins with "This lack of reduction in $EBC_{WB}$..."

5. P1, Line 17: If the site is a likely representative location and EBCWB has not decreased, but the other sites have, how is it representative of ineffective controls on wood burning? Are the same management controls applied everywhere?

This site (Magadino-Cadenazzo) represents the environment south of the Alps and is distinct when compared to the monitoring sites on the Swiss plateau. We believe Magadino-Cadenazzo is a very good representative site for such environments.

6. P2, Line 6: Change "fuelled" to "fueled"

Fuelled is the preferred British English spelling which this manuscript has been prepared in so this has not been changed.

7. P7, Line 2: Change "consistently" to "consistent"

Done.

8. P7, Line 8: What period was considered a daily sample? Midnight-midnight?

The daily averages were indeed midnight to midnight. The text now reads: '...to daily resolution (midnight to midnight)...'

9. P7, Line 10: How were EC data applied with this frequency since BC and EC would overlap only on certain days? Was the EC sampling schedule the same at all sites?

Extra text has been added to clarify this point: "When using aethalometer and EC data together, the aethalometer observations were aggregated (as arithmetic means) to daily resolution (midnight to midnight) to ensure the observations spanned the same time period and

duration. Only days with both EC and absorption observations were used for these comparisons because interpolation of the less frequent EC data was not attempted."

The EC sampling was very similar among the sites with minor variations to sampling frequencies over the analysis period.

10. P7, Line 11: How was PM2.5 measured at all of the sites? What sampling frequency?

   This sentence has been changed to make the type of $PM_{2.5}$ observations clear: "Only daily validated data were kept for analysis with most of the observations being sourced from high volume samplers."

11. Page 7, Line 16: Change "measures" to "data"

   Done.

12. Page 8, Line 4: Were any trend analyses performed on the EC data to test whether EC trends generally followed the overall BC trends?

   Trend analysis of EC has not been conducted as part of this work. However, this is a worthy idea for future work but may be challenging due to less frequent observations before about 2010.

13. Page 8, Line 16: Where was this seasonal pattern observed? At all sites? A range of 3 m2/g is very large, can the authors comment on the physical reasons why the MAC would vary this much on a seasonal basis? They also vary considerably from site to site. Were EC data examined on a site by site and seasonal basis? Do they show this much variability?

   The MAC value discussion has been expanded and now includes a supporting figure (Fig. A2 and below) which shows the seasonal MAC values by site and instrument. The text now discusses that the seasonal cycle was observed in all locations: "The range of the seasonality was as low as $0.4\,\mathrm{m^2\,g^{-1}}$ at Zürich-Kaserne, but as high as $2.3\,\mathrm{m^2\,g^{-1}}$ at Magadino-Cadenazzo. The seasonal variation was also accompanied by intra-instrumental variation."

   The MAC value variation may be due to the woodburning emission source being 'switched on and off' during the year which would substantially alter the seasonal aerosol composition. However, this is not explicitly investigated here and do not wish to speculate on the causes. We point to Zotter et al. 2007 (https://www.atmos-chem-phys.net/17/4229/2017/) where changes in MAC values are discussed.

   The EC observations were checked for seasonality and their range was not as great as 3

units, but showed substantial variation with winter being the most polluted season due to a combination of emission strength and meteorological controls.

[Figure]

Figure 1: Mean seasonal and instrumental mass absorption cross section/mass absorption coefficients (MAC) for 950 nm for six equivalent black carbon (EBC) monitoring sites in Switzerland between 2008 and 2018.

14. Page 9, Line 6: Can the authors provide a brief explanation on how Angstrom exponents were calculated? Are they ratio of two wavelengths or a fit of all wavelengths?

A sentence has been added explaining the calculation process: "Here, $\alpha$ was calculated by curve fitting all absorption wavelengths (370, 470, 520, 590, 660, 880, and 950 nm) with exponential regression models."

15. Page 10, Line 2, Figure 4 caption. Can the authors add the wavelength range for the Angstrom exponents?

The caption of the figure now reads: "Counts of binned Ångström exponents ($\alpha$) for three equivalent black carbon (EBC) monitoring sites' hourly absorption observations in Switzerland between 2014 and 2018. $\alpha$ has been calculated by seven wavelengths between 370 and 950 nm."

16. Page 12, Line 4: What about wood burning as local emissions?

This monitoring site would be influenced by local woodburning and other emissions, but based on the data, this influence is very small. The sentence states this clearly: "This monitoring site is located at an elevation of 1031 metres, is isolated from significant local emissions, and is intermittently in the boundary layer and therefore, at times is not influenced by surface source activities."

17. Page 12, Line 9: Why would this feature be stronger for traffic than wood burning sources?

Although this feature is discussed later in the manuscript, this sentence has been expanded: " This feature was clearer for $EBC_{TR}$ than $EBC_{WB}$ suggesting traffic, rather than woodburning emissions are more influential at Rigi-Seebodenalp."

18. Page 14: Line 23: It appears from the figures that a couple of sites do not have data before 2013. Trend analyses performed over this short of period can be misleading when compared to sites with longer periods.

The EBC monitoring sites do have different start dates which creates a situation where the trend analysis is potentially weakened, these are the data which exist and little can be done to address this.

19. Page 14: Line 26: The Payerne site appears to have a break around 2014, after which the data are relatively flat. A similar pattern may occur at Magadin-Cadenazzo and Zurich sites. Is this an instrument artefact?

We have investigated these potential breakpoints thoroughly. We have added a supplementary figure (Fig. A4 and below) and a paragraph in the trends section explaining these instrument artefacts. The paragraph reads:
"$EBC_{TR}$ trends at Payerne, Magadino-Cadenazzo, and potentially Zürich-Kaserne suggest that breakpoints in the observations are present at the middle of 2013 (Fig. 9). Curiously, these changes were not observable in the absorption observations themselves, but could be detected in the calculated $\alpha$ with adaptive Kolmogorov-Zurbenko filters (KZA filters) (Fig.A4). The times of these small breakpoints could not be robustly traced to operational activities, but they might have been caused by different batches of aethalometer filter-tape which were continuously introduced across the monitoring network at this time. Therefore, these data suggest that these breakpoints are instrument artefacts. The identification of these features reinforces that the use of the aethalometer model is very useful, it is a pragmatic

technique which requires careful evaluation."

[Figure]

Figure 2: Time series of daily and adaptive Kolmogorov-Zurbenko filtered (KZA) Ångström exponents ($\alpha$) for six equivalent black carbon (EBC) monitoring sites' absorption observations in Switzerland between 2008 and 2018. $\alpha$ has been calculated by wavelengths between 370 and 950 nm.

20. Page 15: Line 12-13: But the time period is much shorter and during this period, other sites had flat trends (Payerme and Magadino), so it may not be fully reflective of what is happening over the longer time period.

   We have conducted to the trend analysis on the observations which are available. We accept that the non-uniform data spans is a limitation of the study, but this cannot be changed due to the variations of the observational record and believe the trends are representative of the site types discussed.

21. Page 16, Line 5: Over the same years?

   The EBC/PM$_{2.5}$ ratio analysis is conducted across the same period. The sentence has been modified to make this clear: "The seasonal EBC/PM$_{2.5}$ ratios for the five Swiss EBC monitoring sites between 2014 and 2018 are shown..."

---

## Author Comment (AC2) · 29 Jan 2020

**Author responses to Referee #3's comments of amt-2019-351 (*Evaluation of equivalent black carbon (EBC) source apportionment using observations from Switzerland between 2008 and 2018*)**

January 29, 2020

Stuart K. Grange*, Hanspeter Lötscher, Andrea Fischer, Lukas Emmenegger, and Christoph Hueglin

*stuart.grange@empa.ch

**Response to reviewers**

**Referee #3**

Grange et al. present an evaluation of source apportionment via the aethalometer model from six Swiss sites. They compare their obtained fossil and wood burning fractions from the model to expected diurnal patterns and ambient temperature. They also compare the aerosol Ångstroem exponents (AAE) they use in the model to the binned AAEs of the full (non-apportioned) time series. Finally, they also show and discuss the trends in fossil/wood burning fractions. This article is informative the figures are well designed and illustrative. However, there are some important issues regarding the data processing, at least as it is outlined in the text. I also note that the technical aspects of the paper are not as in depth as work by e.g. (Fuller et al., 2014) or (Zotter et al., 2017) who compare their data to external tracers, such that an experienced user of aethalometer data might learn more from the reported trends and source apportionment than the technical aspects presented. Nevertheless I recommend publication after the following issues are addressed.

Thank you. We have addressed all of the referee's comments below.

**Major issues**

- Most data in the paper are from the Swiss National Air Pollution Monitoring Network (NABEL). I tried to access this site using the link in the references, but found that the link was no longer active, whereas the authors state that data are available as described in the text. Furthermore there is no description of data quality control/assurance. Are all data aggregated to hourly resolution or are some data points removed due to instrumental issues etc.? If points are removed, what is the threshold at which an hourly data point is still reported (i.e. what number of missing points are allowed?). The authors should update the information they

provide on data availability and describe the quality control measures. Payerne and Rigi are EMEP sites and aethalometer data are publicly available at ebas.nilu.no.

We thank the referee for reporting these data accessibility issues. In response to this, we have deposited the absorption observations, site metadata, and instrument locations data used in the data analysis in a persistent and publicly accessible data repository (https://doi.org/10.5281/zenodo.3626658). The data section now contains more information on data quality control and assurance procedures with a new citation which outlines these operations in detail.

- The authors do not seem to have access to time series at the original time resolution of the instruments since they write 'Generally, the observations were stored as hourly means, but for the data which was at higher resolution (10 and 30 minute means)', implying that raw instrument output was not available? Checking these time series is important for several reasons, e.g. to ensure correct application of the automatic loading compensation in the AE33 instruments (i.e. there should be no structure in the absorption or EBC time series related to tape advances), or that no short term spikes are present in the data which could bias the averaging. Plotting the AAE over time is also useful. This lack of information weakens the arguments the authors make regarding the failure of the aethalometer model apparent at the San Vittore site. In the text they assert that this failure is due to the presence of a third source (fresh wood smoke). Here it is important to rule out other causes of failure, which might arise due to errors in the data processing (e.g. incorrect loading compensation). If possible, time series should be shown in the annex. If not, this weakness should be mentioned in the text.

We have conducted our data analysis on observations which have been subjected to quality control and assurance procedures (see the response above) and therefore these data are not "raw" data off the instruments. We would argue that the use of such data is more relevant to the research community because such data are generally what is accessible to the public. We do however accept that using the raw instrument outputs may offer some additional value for testing and checking, it does not weaken our conclusions or results.

We have added a paragraph further detailing the data processing and quality control and assurance procedures. The paragraph reads: "The standard procedure for absorption monitoring data was for 10 and 30 minute means to be calculated from one minute observations

which were logged on-site directly from the instrument. All aggregations required a data capture threshold of 60 % for a valid summary to be produced. Additional quality control and assurance procedures were undertaken quarterly including: cleaning of the inlet, leak testing, and cleaning of the analytical zone of the instruments. The responses of the AE33 instruments were also checked with optical reference filters regularly. The data were ratified on a monthly basis and compared across the different measurement sites along with other air pollutants and suspicious measurements were invalidated. The raw instrument outputs are archived, but are not routinely used in the data processing within the NABEL monitoring network. However, the raw data can be consulted in case of questions concerning data quality."

A time series of $\alpha$ has also been included (Fig. A4 and below) to address this, and other comments.

[Figure]

Figure 1: Time series of daily and adaptive Kolmogorov-Zurbenko filtered (KZA) Ångström exponents ($\alpha$) for six equivalent black carbon (EBC) monitoring sites' absorption observations in Switzerland between 2008 and 2018. $\alpha$ has been calculated by wavelengths between 370 and 950 nm.

**Minor comments**

- Change 'appointment' in the title to 'apportionment'.

  Done.

- Pg 1, ln 16: change 'has be' to 'has been'

  Done.

- Pg 1, ln 17: PM2.5 is undefined

  $PM_{2.5}$ is now defined in the abstract and the first occurrence in text.

- Pg 2, ln 19: Suggest using 'direct legal limits' since in the unlikely event that BC is itself above the limit values for PM2.5 then BC would be above the legal limit. Non-binding provisions for BC limits are also included in the Gothenburg protocol, see also (Shapovalova, 2016).

  The text now reads "...direct legal limits...".

- Pg 5, ln 12: Data from AE31 instruments are included and the authors write 'All absorption observations had been compensated for the filter loading and shadowing effects with the instrument model's respective algorithms before this analysis was undertaken'. However, the output from the AE31 is slightly different to that of the AE33 in that a correction factor for loading is not automatically applied by the instrument and must be done in post processing. Please clarify.

  This text has been altered to clarify the differences between the two instrument models: "The differences between the AE31 and AE33 technologies mean that the algorithms which compensate or correct for filter shadowing effects and filter loading effects are different. Notably, data from AE31 must be compensated for such effects with a post-processing procedure while the AE33's algorithms are conducted on-board as part of the measurement cycle. For the full description of the compensation procedures, see...".

- Pg 7, ln 9: The authors write 'When using aethalometer and EC data together, the aethalometer observations were aggregated (as arithmetic means) to daily 10 resolution'. I assume the data were averaged to the start and end times of the filters? In which case it would be better to write something like 'data were window averaged to the start and end times of the filter sampling'.

This point was also brought up by Referee #2 and has been addressed. The sentence is now much clearer and describes the data handling step in detail: "When using aethalometer and EC data together, the aethalometer observations were aggregated (as arithmetic means) to daily resolution (midnight to midnight) to ensure the observations spanned the same time period and duration. Only days with both EC and absorption observations were used for these comparisons because interpolation of the less frequent EC data was not attempted.".

- Pg 7, ln 23: Use 'recommended by' instead of 'reported by'.

  Done.

- Pg 8, ln 12: What was the least squares algorithm used? Given that there is uncertainty in both x and y, orthogonal distance regression should be used.

  We used simple least squares regression models. Although we accept that uncertainty in the EC observations ($x$) could be argued, the EN16909 thermal method used to determine EC is a reference method and can also be considered "truth". The very high degree of correlation between EBC and absorption would also result in the different estimators producing very similar results. We believe the use of total least squares/orthogonal distance regression is a preference and consider our approach valid for our analysis.

- Pg 8, ln 14: Here it would be better to use 'an average of', rather than a 'range of' since the range is later given as 3 m2/g. i.e. write an average of 10 +- 3 m2/g

  The sentence has been altered and now reads: "The sites' MAC values at 950 nm had an average of $11.3 \pm 2.9 \, \mathrm{m^2 \, g^{-1}}$ during the analysis period..." Please note that the discussion on the MAC values has been expanded due to the comments of Referee #2, point 13.

- Pg 8, ln 18. The report by Anonymous Referee #2 discusses the issue of the large variability in MAC. How much of the variation in MAC is actually due to instrument differences? It is natural to expect some variability due to this. The effect of this will have been somewhat obscured by using the rolling average presented in the paper, so that Fig. A1 likely does not show the full magnitude of the effect. It would be informative to include e.g. seasonal MAC values on an instrument by instrument basis e.g. for each line of Table 2. A version of Fig. 3 highlighting instrument shifts would also be informative.

  A table containing mean MAC values by site, season, and instrument was produced but it is not practical to include in the manuscript due to the length of the table (154 rows).

Therefore, a supplementary figure (Fig. A2 and below) has been added showing these data. Fig. 3 (also below) has also been enhanced to show the times when instruments were changed as requested. Please note that the discussion on the MAC values has been expanded due to the comments of Referee #2, point 13.

[Figure]

Figure 2: Mean seasonal and instrumental mass absorption cross section/mass absorption coefficients (MAC) for 950 nm for six equivalent black carbon (EBC) monitoring sites in Switzerland between 2008 and 2018.

- Pg 8, ln 27: change 'is' to 'are'.

  Done.

- Pg 14, ln 12: change 'negative' to 'negatively'

  Done.

- Pg 22, Figure A2 caption: Please change the caption to reflect that these are the steps used in the paper and not a general requirement, e.g. to 'A flow chart of the data processing steps used to apply the aethalometer model'. Some of the steps could have been done in a different order, and apportionment of absorption coefficients is also possible without conversion to EBC.

[Figure]

Figure 3: Mass absorption cross section (MAC) coefficients for 950 nm for different monitoring sites as calculated by rolling simple linear regression models with windows of 180 days (alignment of the window was the centre of the period).

Done.

**References**

Fuller, G. W., Tremper, A. H., Baker, T. D., Yttri, K. E., and Butterfield, D. J. A.: Contribution of wood burning to PM10 in London, 87, 87-94, 2014.

Shapovalova, D.: The Effectiveness of Current Regulatory Models of Gas Flaring in Light of Black Carbon Emissions Reduction in the Arctic, in: Global Challenges in the Arctic Region: Sovereignty, Environment, and Geopolitical Balance, Routledge, 325-344, 2016.

Zotter, P., Herich, H., Gysel, M., El-Haddad, I., Zhang, Y., Močnik, G., Hüglin, C., Baltensperger, U., Szidat, S., Prévôt, A. S. J. A. Evaluation of the absorption Ångström exponents for traffic and wood burning in the Aethalometer-based source apportionment using radiocarbon measurements of ambient aerosol, 17, 4229-4249, 2017

---

## Referee Report (RR1)

**Review of Evaluation of equivalent black carbon (EBC) source apportionment using observations from Switzerland between 2008 and 2018 by S. Grange et al.**

Grange et al., submit a revised version of their earlier manuscript. I recommend publication pending correction of one major issue and a number of small revisions.

Major issue:

It is well beyond the scope of this article (and journal) to discuss in detail the effectiveness of various emission control technologies, BC abatement strategies, environmental management and the effectiveness of regulations. These important issues are often complex and nuanced, but are discussed in very brief terms in this work.

For example the authors write: 'the diesel market share for passenger vehicles sharply declined in the wake of the Volkswagen emission scandal ("dieselgate") after 2015, and this market change may also be a contributory factor in the EBCTR trends observed after about 2016.' A simpler explanation for reductions in traffic related BC might be the decline in the non-DPF diesel fleet share, and given that in many circumstances that DPF-equipped Diesel passenger cars will emit less PM than gasoline cars, the statement is highly speculative about an important public health topic.

In another example the authors state 'If BC was perfectly controlled at the source of emission, Switzerland could achieve a 6–14 % reduction of PM2.5 concentrations to aid with the compliance with the annual and daily limits set for PM2.5'. It is not clear to me if this should be taken as a suggestion to ignore the issue of BC in Switzerland or focus on it. It is also not feasible. The question of whether to focus on BC in order to reduce total PM would depend on many factors including the marginal abatement costs compared to other sources. Given that the BC fraction is quite low, there is likely to be other sources with lower marginal abatement costs, i.e. 'low hanging fruit', and so this sentence would actually suggest the issue is not important. What about co emissions? Reducing BC emissions would reduce the total PM by more than the BC fraction alone via associated reductions in NOx, VOCs, primary organics etc. Clearly the issue is not as simple as presented here.

In the conclusions the authors state 'This is evidence of ineffective management of reducing BC and PM emissions from domestic wood burning which has been noted in the past'. No citation given, and a potentially defamatory statement.

All such discussion should be removed, especially since doing so would in no way weaken the authors' conclusions regarding aethalometer data treatment.

Minor issues:

The authors use absorption at 950 nm to determine EBC for trends and as one of the wavelength pairs in the aethalometer model. While this is fine (the only advantage of 880 nm I can see is that the error is slightly lower), and is also used by Zotter et al., the absorption in 880 nm channel converted using collocated EC is presented as the 'strict' definition of EBC and is even presented as such in Fig. 1. I suggest to revisit Sect.

1.2 where this is discussed, possibly changing the figure to show either both the more common definition and the definition used in the paper.

On Pg. 9 the authors discuss a decline in the MAC value at Rigi, suggesting this is instrument related. However, there does appear to be a continued decline after the instrument change and so a real effect is possible and might merit further investigation (though not necessarily as a part of this work).

A number of small corrections and suggestions now follow:

Pg. 1, Ln 6: change *BC emitted by different combustion processes have* to *BC emitted by different combustion processes has*

Pg. 3, Ln 4-10: Here it would be natural to introduce the term Ångstroem exponent when discussing spectral dependence of absorption from different sources, and to mention briefly that a major issue with aethalometer model is that it is highly sensitive to this parameter which must be assumed a priori. Another issue is that the choice of wavelength pair influences the model results.

Pg. 4, Ln 2: Change *EBC is only a measurement definition* to *EBC is only an operational definition*

Pg. 4, Ln 10: Change *but* to *it*

Pg. 5, Figure caption suggests that all Swiss EBC monitoring sites are included, but this is not the case since (at least) Jungfraujoch also has EBC measurements, hence the caption overstates how comprehensive the work is. Suggest changing to 'The six Swiss monitoring sites in this study' or similar.

Pg. 9, Ln 3: I think the authors should mention that there is come clear variation in MAC due to instrument changes and that a large part of the standard deviation in MAC is due to this. This is pretty clear from the figure.

Pg. 9, Ln 3 (and elsewhere): Consistently use a space between numbers and units.

Pg. 13, Fig. 6: Caption is too brief to understand the figure on its own. Please indicate in the caption that the figure shows EBC concentrations vs wind speed and direction and that the numbers in the figure are wind speed in m s$^{-1}$.

Pg. 17, Ln 22: Quote marks are not needed, since although the implication seems to be that there are anthropogenic influences on wildfires, they are nevertheless classified as natural. I suggest either stating the issue explicitly e.g. 'from naturally caused combustion processes (which are nevertheless influenced by anthropogenic activity [citation])' or use 'wildfires'.

---

## Author Response (AR2)

**Author responses to referee comments of amt-2019-351 (*Evaluation of equivalent black carbon (EBC) source apportionment using observations from Switzerland between 2008 and 2018*)**

March 15, 2020

Stuart K. Grange*, Hanspeter Lötscher, Andrea Fischer, Lukas Emmenegger, and Christoph Hueglin

*stuart.grange@empa.ch

**Response to editor's comments**

Thank you for addressing the reviewer comments. Please consider the remaining issues as raised by the reviewer. They are rather minor but I completely agree with the concerns. AMT is a measurement techniques journal and measurements are the focus of the manuscript hence there should be no value statements on policy past or present as it is beyond the topic and not really discussed in and supported by the current manuscript. Even the mention of Volkswagen in terms of Diesel problems should be eliminated as in the end of the day, many more manufacturers were found to have manipulated the emission tests. So in a scientific paper Volkswagen should not be singled out.

We have addressed the referee's comments below and have altered the manuscript to remove the interpretation and discussion of the trends observed. The manuscript now presents itself as exclusively a measurement technique paper. Please see our itemized responses below and the accompanying 'diff' file for all the specific changes.

**Response to reviewers**

Grange et al., submit a revised version of their earlier manuscript. I recommend publication pending correction of one major issue and a number of small revisions.

Thank you. All of the issues have been addressed and our comments are below.

**Major issue**

It is well beyond the scope of this article (and journal) to discuss in detail the effectiveness of various emission control technologies, BC abatement strategies, environmental management and

the effectiveness of regulations. These important issues are often complex and nuanced, but are discussed in very brief terms in this work.

For example the authors write: 'the diesel market share for passenger vehicles sharply declined in the wake of the Volkswagen emission scandal ("dieselgate") after 2015, and this market change may also be a contributory factor in the EBCTR trends observed after about 2016.' A simpler explanation for reductions in traffic related BC might be the decline in the non-DPF diesel fleet share, and given that in many circumstances that DPF-equipped Diesel passenger cars will emit less PM than gasoline cars, the statement is highly speculative about an important public health topic.

In another example the authors state 'If BC was perfectly controlled at the source of emission, Switzerland could achieve a 6–14 % reduction of PM2.5 concentrations to aid with the compliance with the annual and daily limits set for PM2.5'. It is not clear to me if this should be taken as a suggestion to ignore the issue of BC in Switzerland or focus on it. It is also not feasible. The question of whether to focus on BC in order to reduce total PM would depend on many factors including the marginal abatement costs compared to other sources. Given that the BC fraction is quite low, there is likely to be other sources with lower marginal abatement costs, i.e. 'low hanging fruit', and so this sentence would actually suggest the issue is not important. What about co emissions? Reducing BC emissions would reduce the total PM by more than the BC fraction alone via associated reductions in NOx, VOCs, primary organics etc. Clearly the issue is not as simple as presented here.

In the conclusions the authors state 'This is evidence of ineffective management of reducing BC and PM emissions from domestic wood burning which has been noted in the past'. No citation given, and a potentially defamatory statement.

All such discussion should be removed, especially since doing so would in no way weaken the authors' conclusions regarding aethalometer data treatment.

We agree, the regulation of emissions is a complex topic and the statements made may have been too simple and not supported well enough. We have removed all statements, discussion, and interpretation of the EBC trends. The abstract, Section 3.5, Section 3.6, and the conclusion sections have been altered. Please see the 'diff' file for the specific changes. We too believe, these changes do not weaken the article in respect to the use of the aethalometer model.

**Minor issues**

- The authors use absorption at 950 nm to determine EBC for trends and as one of the wavelength pairs in the aethalometer model. While this is fine (the only advantage of 880 nm I can see is that the error is slightly lower), and is also used by Zotter et al., the absorption in 880 nm channel converted using collocated EC is presented as the 'strict' definition of EBC and is even presented as such in Fig. 1. I suggest to revisit Sect. 1.2 where this is discussed, possibly changing the figure to show either both the more common definition and the definition used in the paper.

  We indeed used the 950 nm variable for the IR input to the aethalometer model as recommended by Zotter et al. (2017) who evaluated the sensitivity of the outcome of the aethalometer model to different combinations of wavelengths. We have enhanced Fig. 1 (below) to indicate the two wavelengths used in the aethalometer models. We have also added additional text explaining the use of the different wavelengths and the figure's new annotations.

- On Pg. 9 the authors discuss a decline in the MAC value at Rigi, suggesting this is instrument related. However, there does appear to be a continued decline after the instrument change and so a real effect is possible and might merit further investigation (though not necessarily as a part of this work).

  This section has been reworded to reflect this idea:
  "This instrument-derived feature was only uncovered due to the presence of collocated EC observations, but due to a lack of additional data, we do not wish to speculate further on what was the cause for this decline in the MAC coefficient and may warrant further investigation elsewhere".

A number of small corrections and suggestions now follow:

- Pg. 1, Ln 6: change BC emitted by different combustion processes have to BC emitted by different combustion processes has

  Done.

- Pg. 3, Ln 4-10: Here it would be natural to introduce the term Ångström exponent when discussing spectral dependence of absorption from different sources, and to mention briefly

[Figure]

Figure 1: Demonstration of different aethalometer absorption dependence on wavelengths for two monitoring sites in Switzerland with distinct aerosol characteristics. Data have been filtered to a single observation (hourly) to show the dependence on the dominating sources and the 470, 880, and 950 nm channels are highlighted and explained in text. Magadino-Cadenazzo has a stronger absorption in the UV region due to woodburning emissions.

that a major issue with aethalometer model is that it is highly sensitive to this parameter which must be assumed *a priori*. Another issue is that the choice of wavelength pair influences the model results.

The introduction of Ångström exponents now appears in this subsection. Explanation of the wavelength pairs used is now more explicit and mentioning that the Ångström exponents must be chosen on a case by case basis has now been included:

"The spectral dependence of the light absorption can be described by: $b_{abs} \propto \lambda^{-\alpha}$ where $b_{abs}$ is the absorption, $\lambda$ is the wavelength, and $\alpha$ is the Ångström exponent. $\alpha$ is the exponential slope of a regression model for absorption as a function of wavelength. $\alpha$ for woodburning sourced BC results in higher values due to the increased absorption in the UV region when compared to BC emitted by vehicular sources (Fig. 1). "

"... UV and IR $\alpha$ values are also required for the source apportionment and need to be carefully chosen on a case-by-case basis however. "

- Pg. 4, Ln 2: Change EBC is only a measurement definition to EBC is only an operational definition

  Done.

- Pg. 5, Figure caption suggests that all Swiss EBC monitoring sites are included, but this is not the case since (at least) Jungfraujoch also has EBC measurements, hence the caption overstates how comprehensive the work is. Suggest changing to 'The six Swiss monitoring sites in this study' or similar.

  The caption now reads:

  "The six Swiss equivalent black carbon (EBC) monitoring sites used in the analysis and their site classifications..."

- Pg. 9, Ln 3: I think the authors should mention that there is come clear variation in MAC due to instrument changes and that a large part of the standard deviation in MAC is due to this. This is pretty clear from the figure.

  The instrument cycling is discussed in the sentence and is shown in Fig. 3 and Fig. A2.

- Pg. 9, Ln 3 (and elsewhere): Consistently use a space between numbers and units.

  Done, all number-unit combinations have been checked for consistency.

- Pg. 13, Fig. 6: Caption is too brief to understand the figure on its own. Please indicate in the caption that the figure shows EBC concentrations vs wind speed and direction and that the numbers in the figure are wind speed in m s-1.

  The caption now reads:

  "Mean polar plots of equivalent black carbon (EBC) components for Magadino-Cadenazzo between 2011 and 2018. Fill-colours show EBC concentrations by wind speed (in $\mathrm{m\,s^{-1}}$) and wind direction."

- Pg. 17, Ln 22: Quote marks are not needed, since although the implication seems to be that there are anthropogenic influences on wildfires, they are nevertheless classified as natural. I suggest either stating the issue explicitly e.g. 'from naturally caused combustion processes (which are nevertheless influenced by anthropogenic activity [citation])' or use 'wildfires'.

  The sentence has been reworded and the quote marks have been removed:

  "BC is almost exclusively an anthropogenic pollutant with generally minor and intermittent contributions from wildfires.".

**Other changes**

The figures have been altered very slightly to make the title "strips" narrower for better presentation.

[revised manuscript text omitted]